# Deciphering anomalous heterogeneous intracellular transport with neural networks

Daniel Han[1,2,3]*, Nickolay Korabel[1], Runze Chen[4], Mark Johnston[2], Anna Gavrilova[1,2], Victoria J Allan[2]*, Sergei Fedotov[1]*, Thomas A Waigh[3,5]*

[1]Department of Mathematics, University of Manchester, Manchester, United Kingdom; [2]School of Biological Sciences, University of Manchester, Manchester, United Kingdom; [3]Department of Physics and Astronomy, University of Manchester, Manchester, United Kingdom; [4]Department of Computer Science, University of Manchester, Manchester, United Kingdom; [5]The Photon Science Institute, University of Manchester, Manchester, United Kingdom

**Abstract** Intracellular transport is predominantly heterogeneous in both time and space, exhibiting varying non-Brownian behavior. Characterization of this movement through averaging methods over an ensemble of trajectories or over the course of a single trajectory often fails to capture this heterogeneity. Here, we developed a deep learning feedforward neural network trained on fractional Brownian motion, providing a novel, accurate and efficient method for resolving heterogeneous behavior of intracellular transport in space and time. The neural network requires significantly fewer data points compared to established methods. This enables robust estimation of Hurst exponents for very short time series data, making possible direct, dynamic segmentation and analysis of experimental tracks of rapidly moving cellular structures such as endosomes and lysosomes. By using this analysis, fractional Brownian motion with a stochastic Hurst exponent was used to interpret, for the first time, anomalous intracellular dynamics, revealing unexpected differences in behavior between closely related endocytic organelles.

*For correspondence:
daniel.han@manchester.ac.uk (DH);
viki.allan@manchester.ac.uk (VJA);
sergei.fedotov@manchester.ac.uk (SF);
t.a.waigh@manchester.ac.uk (TAW)

Competing interests: The authors declare that no competing interests exist.

## Introduction

The majority of transport inside cells on the mesoscale (nm-100μm) is now known to exhibit non-Brownian anomalous behavior (*Metzler and Klafter, 2004*; *Barkai et al., 2012*; *Waigh, 2014*). This has wide ranging implications for most of the biochemical reactions inside cells and thus cellular physiology. It is vitally important to be able to quantitatively characterize the dynamics of organelles and cellular responses to different biological conditions (*van Bergeijk et al., 2015*; *Patwardhan et al., 2017*; *Moutaux et al., 2018*). Classification of different non-Brownian dynamic behaviors at various time scales has been crucial to the analysis of intracellular dynamics (*Fedotov et al., 2018*; *Bressloff and Newby, 2013*), protein crowding in the cell (*Banks and Fradin, 2005*; *Weiss et al., 2004*), microrheology (*Waigh, 2005*; *Waigh, 2016*), entangled actin networks (*Amblard et al., 1996*), and the movement of lysosomes (*Ba et al., 2018*) and endosomes (*Flores-Rodriguez et al., 2011*). Anomalous transport is currently analyzed by statistical averaging methods and this has been a barrier to understanding the nature of its heterogeneity.

Spatiotemporal analysis of intracellular dynamics is often performed by acquiring and tracking microscopy movies of fluorescing membrane-bound organelles in a cell (*Rogers et al., 2007*; *Flores-Rodriguez et al., 2011*; *Chenouard et al., 2014*; *Zajac et al., 2013*). These tracks are then commonly interpreted using statistical tools such as the mean square displacement (MSD) averaged over the ensemble of tracks, $\langle \Delta r^2(t) \rangle$. The MSD is a measure that is widely used in physics, chemistry and

biology. In particular, MSDs serve to distinguish between anomalous and normal diffusion at different temporal scales by determining the anomalous exponent $\alpha$ through $\langle \Delta r^2(t) \rangle \sim t^\alpha$ (**Metzler and Klafter, 2000**). Diffusion is defined as $\alpha = 1$, sub-diffusion $0 < \alpha < 1$ and super-diffusion $1 < \alpha < 2$ (**Klafter and Sokolov, 2011**). To improve the statistics of MSDs, they are often averaged over different temporal scales, forming the time-averaged MSD (TAMSD), $\overline{\Delta r^2(\tau)} \sim \tau^\alpha$, where $\tau$ is the lag time (**Sokolov, 2012**).

For stochastic processes with long-range time dependence such as fractional Brownian motion (fBm), other statistical averaging methods exist. For fBm, the MSD is $\langle B_H^2(t) \rangle \sim t^{2H}$ with the Hurst exponent, $H$ varying between 0 and 1. One can use rescaled and sequential range analysis (**Samorodnitsky, 2016**; **Peters, 1994**) to estimate $H$. The advantage of modeling intracellular transport with fBm is that both sub-diffusion ($0 < H < 1/2$) and super-diffusion ($1/2 < H < 1$) can be explained in a unified manner using only the Hurst exponent. The essence of fBm is that long-range correlations result in random trajectories that are anti-persistent ($0 < H < 1/2$) or persistent ($1/2 < H < 1$). How can we understand persistence in the context of intracellular transport? The term persistence can be understood as the processive motor-protein transport of cargo in one direction, whether it be retrograde or anterograde. From a probabilistic viewpoint, persistence can be interpreted as the cargo being more likely to keep the same direction given it had been moving in this fashion before. Conversely, anti-persistence is interpreted as cargo being more likely to change its direction given it had been moving in that direction before. Anti-persistence can arise if cargo is confined to a local volume in the cytoplasm simply due to crowding or tethering biochemical interactions (**Harrison et al., 2013**), which in effect leads to sub-diffusion (**Weiss et al., 2004**; **Ernst et al., 2012**). By interpreting intracellular cargo transport as fBm, there are two main advantages: we can describe movement with the intuitive biological concepts of persistence and anti-persistence; and we can provide an immediate link to anomalous diffusion since $\alpha = 2H$ for constant H.

Cargo movement in vivo often exhibits random switching between persistent and anti-persistent movement, even in a single trajectory (**Chen et al., 2015**). Therefore, we can model this by a stochastic local Hurst exponent, $H(t)$, which jumps between persistent ($1/2 < H(t) < 1$) and anti-persistent ($0 < H(t) < 1/2$) states. Still, a major challenge exists: how can we estimate a local stochastic Hurst exponent from a trajectory?

Whilst exponent estimation using neural networks is an emerging field (**Bondarenko et al., 2016**), segmentation of single trajectories into persistent and anti-persistent sections based on instantaneous dynamic behavior has not been studied. Instead, hidden Markov models (**Monnier et al., 2015**; **Persson et al., 2013**) and windowed analyses (**Getz and Saltz, 2008**) are commonly used to segment local behavior along single trajectories (see Appendix A for comparisons). Even so, most methods neglect the microscopic processes which are often a feature of intracellular transport (e.g. alternation between 'runs' and 'rests') (**Weiss et al., 2004**; **Chen et al., 2015**; **Fedotov et al., 2018**) and the non-Markovian nature of their motion (**Fuliński, 2017**). fBm was chosen due to its self-similar properties that allow direct analysis at short time scales given by experimental systems; and the experimental evidence for fBm in the crowded cytoplasm (**Weiss et al., 2004**; **Szymanski and Weiss, 2009**; **Krapf et al., 2019**). Moreover, other anomalous diffusion models, such as scaled Brownian motion (**Lim and Muniandy, 2002**), subdiffusive continuous time random walks (**Sokolov, 2012**) and superdiffusive Lévy walks (**Fedotov et al., 2018**) are not suitable to interpret anomalous trajectories on the microscopic level.

Here, we present a new method for characterizing anomalous transport inside cells based on a Deep Learning Feedforward Neural Network (DLFNN) that is trained on fBm. Neural networks are becoming a general tool in a wide range of fields, such as single-cell transcriptomics (**Deng et al., 2019**) and protein folding (**Evans et al., 2018**). We find the neural network is a much more sensitive method to characterise fBm than previous statistical tools, since it is an intrinsically non-linear regression method that accounts for correlated time series. In addition, it can estimate the Hurst exponent using as few as seven consecutive time points with good accuracy.

To test the ability of the DLFNN to segment real-world biological motility, we focused on organelles in the endocytic pathway. This pathway is essential for cell homeostasis, allowing nutrient uptake, the turnover of plasma membrane components, and uptake of growth factor receptors bound to their ligands. Early endosomes then sort components destined for degradation from material that needs to be recycled back to the cell surface or to the Trans-Golgi Network (TGN)

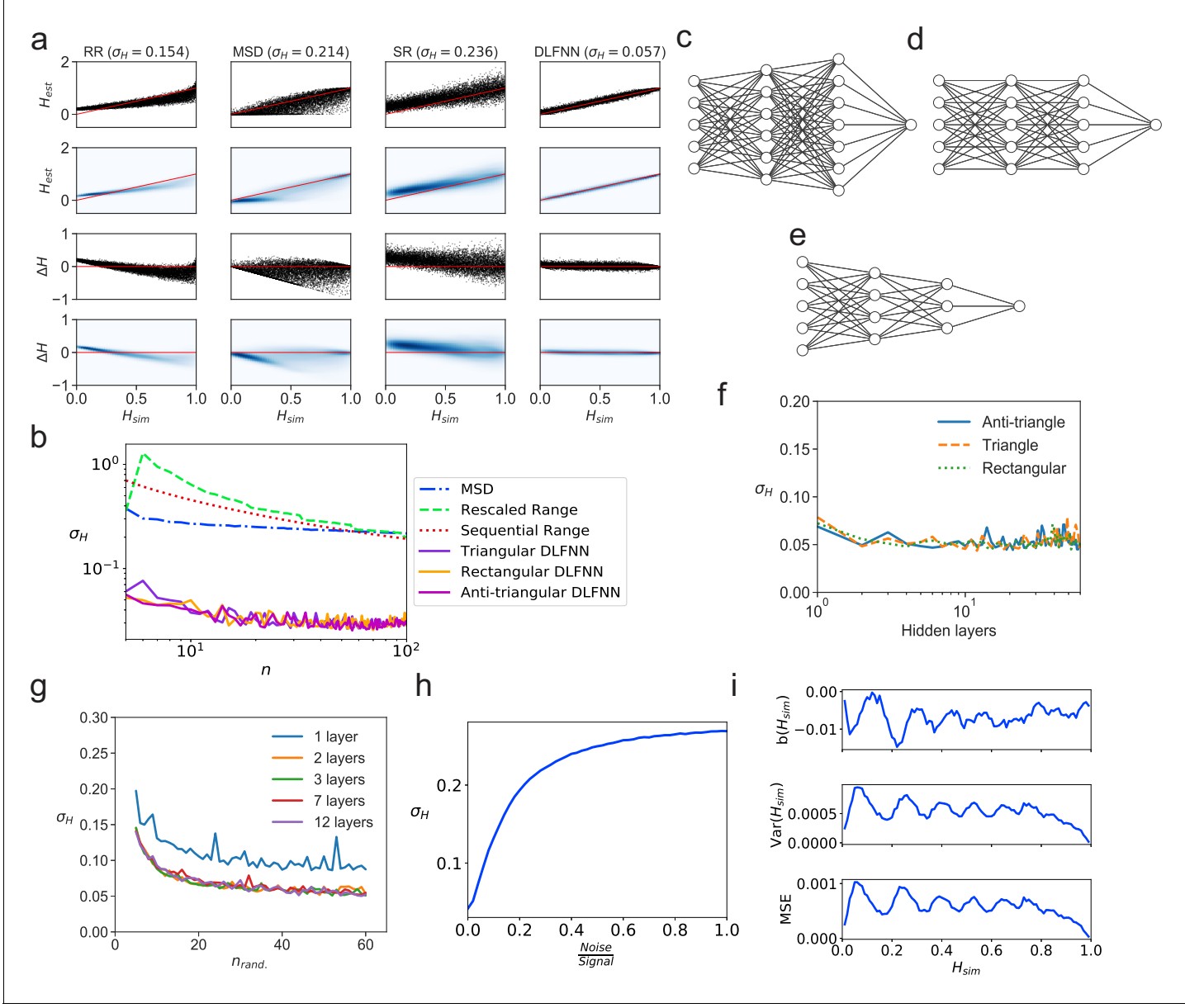

**Figure 1.** Tests of exponent estimation for the DLFNN using $N = 10^4$ simulated fBm trajectories. (a) Plots showing the Hurst exponent estimates of fBm trajectories with $n = 10^2$ data points by a triangular DLFNN with three hidden layers compared with conventional methods. Plots are vertically grouped by Hurst exponent estimation method: (left to right) rescaled range, MSD, sequential range and DLFNN. $\sigma_H$ values are shown in the title. *Top row*: Scatter plots of estimated Hurst exponents $H_{est}$ and the true value of Hurst exponents from simulation $H_{sim}$. The red line shows perfect estimation. *Second row*: Due to the density of points, a Gaussian kernel density estimation was made of the plots in the top row (see Materials and methods). *Third row*: Scatter plots of the difference between the true value of Hurst exponents from simulation and estimated Hurst exponent $\Delta H = H_{sim} - H_{est}$. *Last row*: Gaussian kernel density estimation of the plots in the third row. (b) $\sigma_H$ as a function of the number of consecutive fBm trajectory data points $n$ for different methods of exponent estimation. Example structures for two hidden layers and $n = 5$ time series input points of the anti-triangular, rectangular and triangular DLFNN are shown in (c, d and e), respectively. (f) $\sigma_H$ as a function of the number of hidden layers in the DLFNN for triangular, rectangular and anti-triangular structures. (g) $\sigma_H$ as a function of the number of randomly sampled fBm trajectory data points $n_{rand}$ with different number of hidden layers in the DLFNN shown in the legend. (h) $\sigma_H$ as a function of the noise-to-signal ratio ($\frac{Noise}{Signal}$) (NSR) from Gaussian random numbers added to all $n = 10^2$ data points in simulated fBm trajectories. (i) Plots of bias $b(H_{sim})$, variance $\mathrm{Var}(H_{sim})$ and mean square error (MSE) as functions of $H_{sim}$. For each value of $H_{sim}$, fBm trajectories with $n = 100$ points were simulated and estimated by a triangular DLFNN.

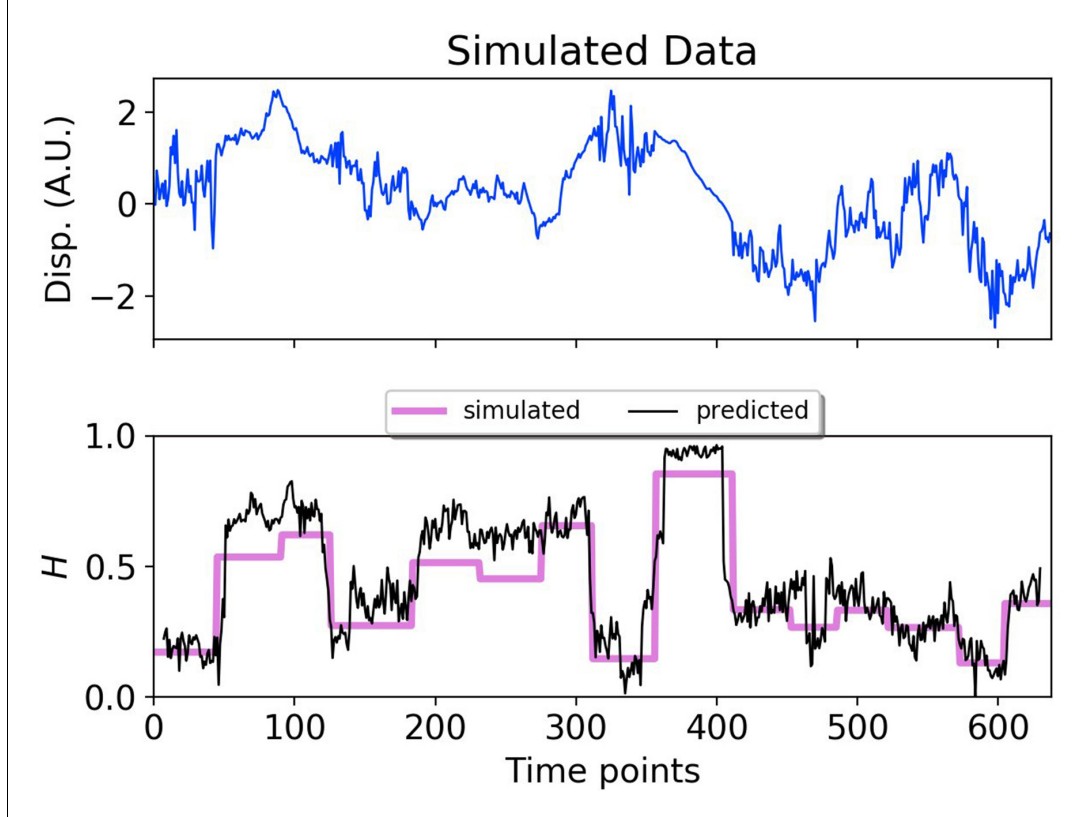

**Figure 2.** DLFNN analysis of a simulated trajectory. *Top*: Plot of displacement as a function of time from a simulated fBm trajectory (blue) with multiple exponent values. *Bottom*: Hurst exponent values used for simulation (magenta), and the DLFNN exponent predictions of the neural network using a 15 point moving window (black).

(*Naslavsky and Caplan, 2018*). Many aspects of endosome function are regulated by Rab5, a small GTPase that is localized to the cytosolic face of early endosomes (*Stenmark and Olkkonen, 2001*). Sorting nexin 1 (SNX1) also localises to early endosomes, where it works with the retromer complex to retrieve and recycle cargoes from early endosomes to the TGN (*Simonetti and Cullen, 2019*). SNX1 achieves this through regulating tubular membrane elements on early endosomes by associating with regions of high membrane curvature (*Carlton et al., 2004*). Early endosomes mature into late endosomes, which then fuse with lysosomes, delivering their contents for degradation (*Huotari and Helenius, 2011*). Endocytic pathway components are highly dynamic, with microtubule motors driving long-distance movement while short-range dynamics involve actin-based motility (*Granger et al., 2014*; *Cabukusta and Neefjes, 2018*), making them ideal test cases for DLFNN analysis. The new method enables the interpretation of experimental trajectories of lysosomes and endosomes as fBm with stochastic local Hurst exponent, H (t). This in turn allows us to unambiguously and directly classify endosomes and lysosomes to be in anti-persistent or persistent states of motion at different times. From experiments, we observe that the time spent within these two states both exhibit truncated heavy-tailed distributions.

To our knowledge, this is the first method which is capable of resolving heterogeneous behavior of anomalous transport in both time and space. We anticipate that this method will be useful in characterizing a wide range of systems that exhibit anomalous heterogeneous transport. We have therefore created a GUI computer application in which the DLFNN is implemented, so that the wider community can conveniently access this analysis method.

## Results and discussion

### The DLFNN is more accurate than established methods

We tested a DLFNN trained on fBm with three hidden layers of densely connected nodes on $N = 10^4$ computer-generated fBm trajectories each with $n = 10^2$ evenly spaced time points and constant Hurst exponent $H_{sim}$, randomly chosen between 0 and 1. The DLFNN estimated the Hurst exponents $H_{est}$ based on the trajectories, and these were compared with those estimated from TAMSD, rescaled range, and sequential range methods (*Figure 1a*). The difference between the simulated and estimated values $\Delta H = H_{sim} - H_{est}$ was much smaller for the DLFNN than for the other methods (*Figure 1a*), and the DLFNN was ~3 times more accurate at estimating Hurst exponents with a mean absolute error $(\sigma_H) \sim 0.05$. Also, the errors in estimation of the DLFNN are more stable across values of $H_{sim}$.

Tracking of intracellular motion usually generates trajectories with a variable number of data points. We therefore compared the performance of the different exponent estimation methods when the number of evenly spaced, consecutive fBm time points in a trajectory varied over $n = 5, 6, ..., 10^2$ points. The DLFNN maintained an accuracy of $\sigma_H \sim 0.05$ across $n$, whereas $\sigma_H$ of other methods increase as $n$ decreases (*Figure 1b*), and was always substantially worse than that of the DLFNN estimation. Different DLFNN structures (see *Figure 1c,d and e*) performed similarly, and introducing more hidden layers did not affect the accuracy of estimation (*Figure 1f and g*). Given that the structure of DLFNN does not significantly affect the accuracy of exponent estimation, a triangular densely connected DLFNN was used for all subsequent analyses.

The structure of a triangular DLFNN means that the input layer consists of $n$ nodes, which are densely connected to $n - 1$ nodes in the first hidden layer, such that at the $l^{th}$ hidden layer, there would be $n - l$ densely connected nodes. Then to estimate the Hurst exponent these nodes are connected to a single node using a Rectified Linear Unit (ReLU) activation function, which returns the exponent estimate. A triangular DLFNN therefore uses only $\sum_{l=0}^{L}(n - l) + 1$ nodes for $L$ hidden layers and $n$ input points, whereas the rectangular structure uses $nL + 1$ nodes and the anti-triangular structure uses $\sum_{l=0}^{L}(n + l) + 1$. The triangular structure results in a significant decrease in training parameters, and hence computational requirements, while maintaining good levels of accuracy. This demonstrates that a computationally inexpensive neural network can accurately estimate exponents.

The DLFNN's estimation capabilities were tested further by inputting $n_{rand}$ randomly sampled time points from the original fBm trajectories. Surprisingly, $\sigma_H \sim 0.05$ is regained even with just 40 out of 100 data points randomly sampled from the time series for any triangular DLFNN with more than one hidden layer (*Figure 1g*). For this method to work with experimental systems, it must estimate Hurst exponents even when the trajectories are noisy. *Figure 1h* shows how the exponent estimation error increases when Gaussian noise with varying strength compared to the original signal is added to the fBm trajectories. Importantly, the DLFNN accuracy $\sigma_H$ at 20% NSR is as good as the accuracy of other methods with no noise (compare 1a and h).

To characterize the accuracy of $H_{sim}$ estimation by the DLFNN, we calculated the bias, $b(H_{sim}) = \mathbb{E}[H_{est}] - H_{sim}$; variance, $\mathrm{Var}(H_{sim}) = \mathbb{E}\left[H_{est} - \mathbb{E}[H_{est}]^2\right]$; and mean square error, $\mathrm{MSE} = \mathrm{Var}(H_{sim}) + b(H_{sim})^2$ (*Figure 1i*). To quantify the efficiency of the estimator the Fisher information of the neural network's estimation needs to be found and the Cramer-Rao lower bound calculated. The values of bias, variance and MSE were very low (*Figure 1i*), which taken together with the simplicity of calculation and the accuracy of estimation even with small number of data points, demonstrates the strength of the DLFNN method. Furthermore, once trained, the model can be saved and reloaded at any time. Saved DLFNN models, code and the DLFNN Exponent Estimator GUI are available to download (see Software and Code).

### DLFNN allows analysis of simulated trajectories with local stochastic Hurst exponents

Estimating local Hurst exponents is fundamentally important because much research has focused on inferring active and passive states of transport within living cells using position-derived quantities such as windowed MSDs, directionality and velocity (*Arcizet et al., 2008*; *Monnier et al., 2015*). The trajectories are then segmented and Hurst exponents measured in an effort to characterize the

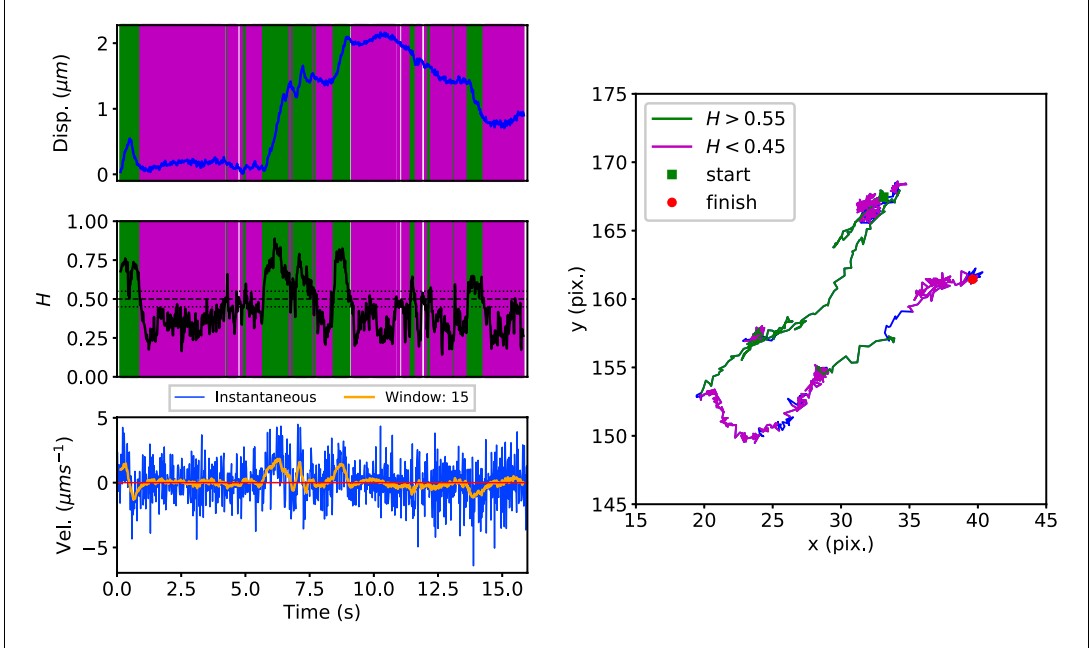

**Figure 3.** DLFNN analysis of a GFP-Rab5 endosome trajectory. *Top*: Plot of displacement from a single trajectory in an MRC-5 cell (blue). Shaded areas show persistent (0.55 < H < 1 in green) and anti-persistent (0 < H < 0.45 in magenta) behaviour. *Middle*: A 15 point moving window DLFNN exponent estimate for the trajectory (black) with a line (dashed) marking diffusion H = 0.5 and two lines (dotted) marking confidence bounds for estimation marking H = 0.45 and 0.55. *Bottom*: Plot of instantaneous and moving (15 point) window velocity. *Right*: Plot of the trajectory with start and finish positions. Persistent (green) and anti-persistent (magenta) segments are shown. For sections that were 0.45 < H < 0.55 were not classified as persistent or anti-persistent and are depicted in blue.

The online version of this article includes the following figure supplement(s) for figure 3:

**Figure supplement 1.** DLFNN analysis of a GFP-SNX1-labeled endosome trajectory, depicted as in *Figure 3*.

**Figure supplement 2.** DLFNN analysis of a lysosome trajectory, depicted as in *Figure 3*.

behavior of different cargo when they are actively transported by motor proteins (*Chen et al., 2015*; *Fedotov et al., 2018*) or sub-diffusing in the cytoplasm (*Jeon et al., 2011*). However, conventional methods such as the MSD and TAMSD need trajectories with many time points ($n \sim 10^2 - 10^3$) to calculate a single Hurst exponent value with high fidelity. In contrast, the DLFNN enables the Hurst exponent to be estimated, directly from positional data, for a small number of points. Furthermore, the DLFNN measures local Hurst exponents without averaging over time points and is able to characterize particle trajectories that may exhibit multi-fractional, heterogeneous dynamics.

To provide a synthetic data set that mimics particle motion in cells, we simulated fBm trajectories with Hurst exponents that varied in time, and applied a symmetric moving window to estimate the Hurst exponent using a small number of data points before and after each time point (*Figure 2*). The DLFNN was able to identify segments with different exponents, and provided a good running estimation of the Hurst exponent values. The DLFNN could also handle trajectories with different diffusion coefficients, and generally performed better than MSD analysis when a sliding window was used (see Appendix B).

## DLFNN analysis reveals differences in motile behavior of organelles in the endocytic pathway

Early endosomes labeled with green fluorescent protein (GFP)-Rab5 undergo bursts of rapid cytoplasmic dynein-driven motility interspersed with periods of rest (*Flores-Rodriguez et al., 2011*; *Zajac et al., 2013*). We therefore applied the DLFNN method to experimental trajectories obtained from automated tracking (*Newby et al., 2018*) data of GFP-Rab5-labeled endosomes in an MRC-5 cell line that stably expressed GFP-Rab5 at low levels (*Figure 3*). A moving window of 15 points identified persistent (green) and anti-persistent (magenta) segments, which corresponded well to

the moving window velocity plots (*Figure 3*, lower panel), confirming that the neural network is indeed distinguishing passive states from active transport states with non-zero average velocity. We then used it to analyze the motility of two other endocytic compartments: SNX1-positive endosomes (*Allison et al., 2017*; *Hunt et al., 2013*) and lysosomes (*Cabukusta and Neefjes, 2018*; *Hendricks et al., 2010*). It successfully segmented tracks of GFP-SNX1 endosomes (*Figure 3—figure supplement 1*) in a stable MRC-5 cell line (*Allison et al., 2017*) and lysosomes visualized using lysobrite dye (*Figure 3—figure supplement 2*). A total of 63–71 MRC-5 cells were analyzed, giving 40,800 (GFP-Rab5 endosome), 11,273 (GFP-SNX1 endosome) and 38,039 (lysosome) tracks that were segmented into 277,926 (GFP-Rab5), 215,087 (GFP-SNX1) and 474,473 (lysosome) persistent or anti-persistent sections, each yielding a displacement, duration and average $H$.

These data revealed intriguing similarities and differences in behavior between the three endocytic components. Analysis of the duration and displacement of segments (Appendix C) revealed that all organelles spent longer in anti-persistent than persistent states (*Figure 4*) but moved much further when persistent (*Appendix 3—figure 1*), as expected. However, GFP-SNX1 endosomes spent much less time than GFP-Rab5 endosomes or lysosomes in an anti-persistent state (*Figure 4*). This difference in behavior was also seen when histograms of the Hurst exponents were plotted (*Figure 5*), as SNX1 endosomes were much less likely to exhibit anti-persistent behavior, particularly with $H<0.3$, than Rab5 endosomes or lysosomes. This was confirmed by fitting the histograms of the Hurst exponent with a six component Gaussian mixture model (*Figure 5b–d*; Appendix D). In contrast, all three organelle classes exhibited a similar range of Hurst exponents when they underwent directionally persistent motion.

To understand organelle motility in the context of cell behavior, an additional layer of complexity needs to be considered - the location of the moving structure within the cell itself. Such information would reveal zones that favor anti-persistent or persistent movement (*Bálint et al., 2013*). Using the neural network, trajectories of GFP-Rab5, GFP-SNX1 endosomes and lysosomes from MRC-5 cells were plotted with colors depicting the changing Hurst exponent at different points in each trajectory (*Figure 6*). For Rab5- and SNX1-positive endosomes, anti-persistent organelles were enriched in the cell periphery, but occasionally underwent long-range persistent movement towards the nucleus (*Figure 6—video 1*; *Figure 6—video 2*), as expected (*Flores-Rodriguez et al., 2011*; *Zajac et al., 2013*; *Hunt et al., 2013*; *Allison et al., 2017*). Lysosomes displayed completely different behavior, with most trajectories being anti-persistent, while the persistent trajectories were not obviously organized spatially (*Figure 6*; *Figure 6—video 3*). The location information together with classification of anti-persistent and persistent trajectories qualitatively shows the regions of high motor-driven activity within the cell for different endocytic organelles.

Many cargos that move along microtubules can switch their direction of motility, between dynein-driven inward (retrograde) motion toward the microtubule minus ends at the cell centre and plus-end-directed outward (anterograde) movement driven by kinesin family members (*Hancock, 2014*). To investigate the characteristics of anterograde and retrograde motility of endocytic organelles, we adapted our method to subdivide persistent segments according to whether the movement occurred towards or away from the user-defined centrosomal region (see Materials and methods). Only tracks with displacement of >0.5μm from their start point were selected, which yielded 2369 Rab5, 2099 SNX1 and 7645 lysosome persistent segments that were then analyzed to give the duration, displacement and velocity of anterograde and retrograde excursions (*Figure 7*; *Table 1*). The anti-persistent segments contained within these tracks were also analyzed.

These statistics revealed that each endocytic organelle moved with different characteristics. GFP-Rab5 endosomes moved much faster than GFP-SNX1 endosomes or lysosomes, particularly in the retrograde direction (*Figure 7*, upper panel). Strikingly, although the GFP-SNX1 endosomes were slowest in both directions, they moved furthest and for longest in each segment, in keeping with the longer duration of persistent segments seen in the global analysis of tracks (*Figure 4*) and higher H values (*Figure 5*). The differences in behavior between Rab5 and SNX1 endosomes is intriguing, since both are recruited to the early endosome by the lipid phosphoinositol-3-phosphate (*Christoforidis et al., 1999*; *Carlton et al., 2004*; *Behnia and Munro, 2005*; *Huotari and Helenius, 2011*). However, SNX1 also senses membrane curvature (*Carlton et al., 2004*), and immunofluorescence labeling of MRC-5 cells with antibodies to Rab5 and SNX1 demonstrated that they reside on distinct domains of larger early endosomes (*Figure 6—figure supplement 1*), as expected *van Weering et al. (2012)*. In addition, while SNX1 endosomes were usually Rab5-positive, there

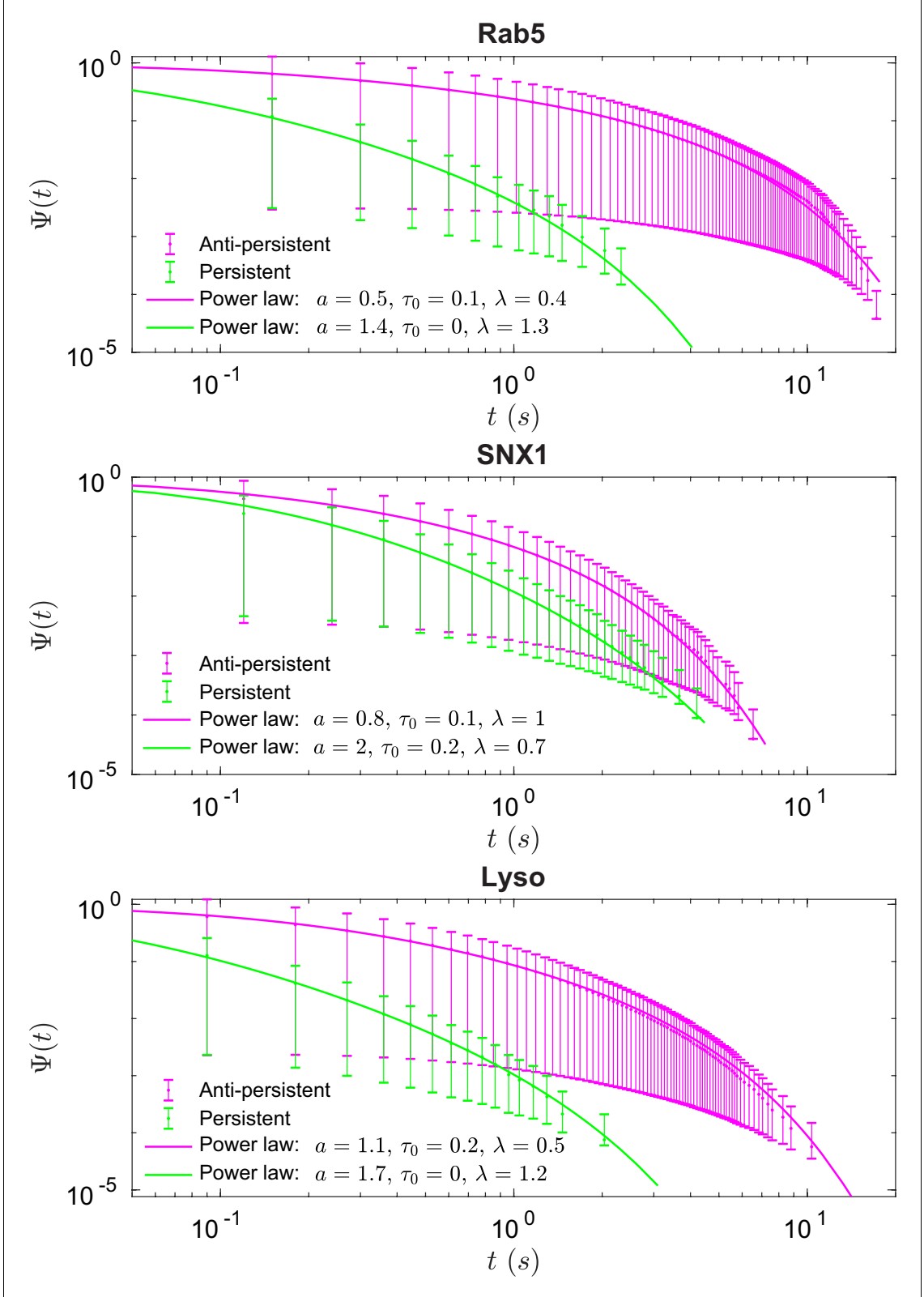

**Figure 4.** Survival functions plotted with error bars for persistent and anti-persistent segments for Rab5-positive endosomes, SNX1-positive endosomes and lysosomes with the power-law fits. Fit parameters can be found in *Appendix 3—table 1*.

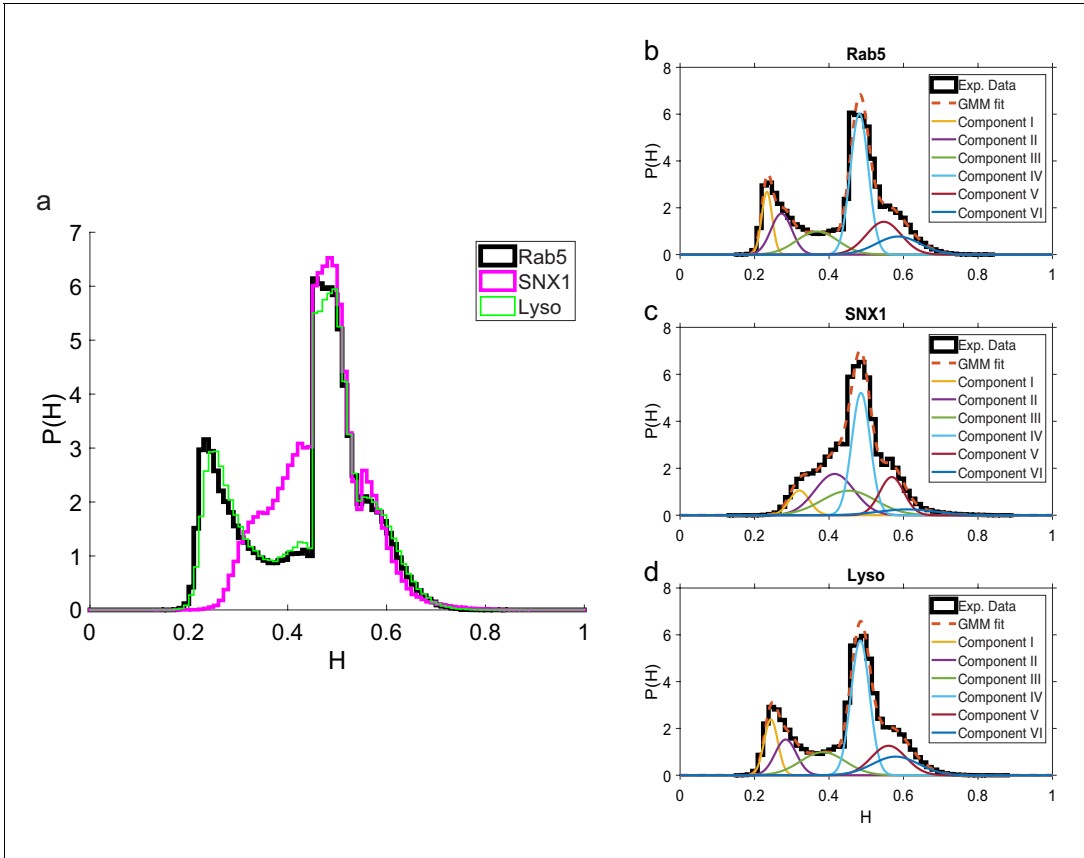

**Figure 5.** Comparison of Hurst exponent distributions for GFP-Rab5, GFP-SNX1 and lysosomes. (**a**) Histograms of Hurst exponents for GFP-Rab5 (black), GFP-SNX1 (magenta) endosomes and lysosomes (green) plot on the same axes for comparison. The individual histograms of Hurst exponents (black solid) for GFP-Rab5-tagged endosomes, GFP-SNX1-tagged endosomes and lysosomes are shown in (**b**, **c** and **d**) respectively. For each histogram, the Gaussian mixture model fit for six components (red dashed) and individual Gaussian distribution components are shown on the same plot. The number of components were chosen through the Bayes information criterion shown in *Appendix 4—figure 1*.

was a significant population of Rab5 endosomes that lacked SNX1, especially smaller early endosomes that were often located in the cell periphery. It is likely that this population of Rab5-positive, SNX1-negative endosomes is particularly motile. The high retrograde velocity of these endosomes might be explained by the recruitment of dynein to Rab5 endosomes via Hook family members (*Bielska et al., 2014*; *Zhang et al., 2014*; *Schroeder and Vale, 2016*; *Guo et al., 2016*). These dynein adaptors have the intriguing property of recruiting two dyneins per dynactin (*Urnavicius et al., 2018*; *Grotjahn et al., 2018*), leading to faster rates of movement in motility assays using purified protein than adaptors that only recruit one dynein per dynactin. Perhaps, SNX1 endosomes move more slowly than Rab5 endosomes because they use a 'single-dynein' adaptor. An alternative explanation could be that SNX1 endosomes are slowed down by interactions with the actin cytoskeleton, since SNX1 domains are enriched in the WASH complex, which in turn controls localized actin assembly (*Gomez and Billadeau, 2009*; *Simonetti and Cullen, 2019*). Actin might also contribute to the slow, steady motion of SNX1 endosomes via myosin motors or the formation of actin comets (*Simonetti and Cullen, 2019*). These interesting possibilities remain to be tested experimentally.

The analysis of anterograde and retrograde segments revealed that lysosomes moved at moderate speed, and were equally fast in both directions, but each burst of movement was short (*Figure 7*, upper panels). In addition, pauses were $\geq 4$ times longer for lysosomes than either early endosome type (*Figure 7*, lower panels). Lysosomes also often changed direction of movement (e.g. *Figure 3— figure supplement 2*), as previously reported (*Hendricks et al., 2010*). So far, no activating dynein adaptor has been identified on lysosomes (*Reck-Peterson et al., 2018*), although several potential

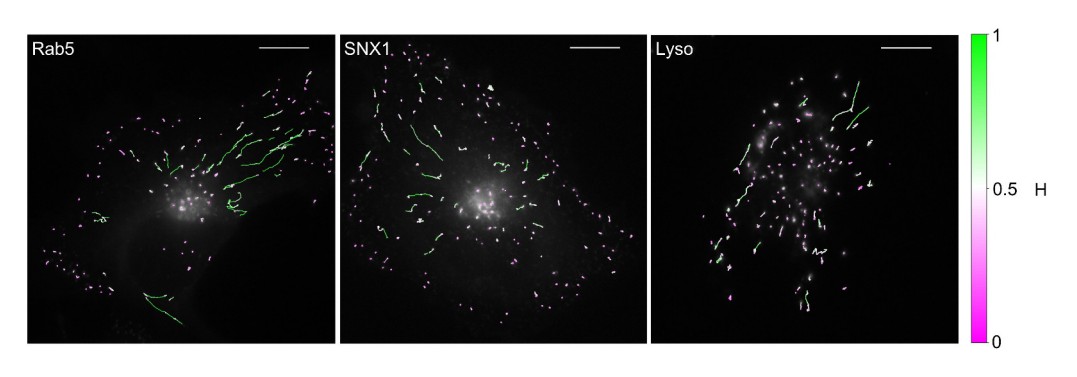

**Figure 6.** MRC-5 cells stably expressing GFP-Rab5, GFP-SNX1 or stained with Lysobrite with tracking data overlaid. The colours show the value of $H$ estimated by the neural network using a 15 point window. The scalebar is 10 µm.

The online version of this article includes the following video and figure supplement(s) for figure 6:

**Figure supplement 1.** Distribution of endogenous Rab5 and SNX1.

**Figure 6—video 1.** Video of MRC-5 cell stably expressing GFP-Rab5.

https://elifesciences.org/articles/52224#fig6video1

**Figure 6—video 2.** Video of MRC-5 cell stably expressing GFP-SNX1.

https://elifesciences.org/articles/52224#fig6video2

**Figure 6—video 3.** Video of MRC-5 cell stained with Lysobrite.

https://elifesciences.org/articles/52224#fig6video3

dynein interactors have been identified, such as RILP (Rab7 interacting lysosomal protein (*Cabukusta and Neefjes, 2018*). Whether this underlies the difference in motile behavior between lysosomes and early endosomes remains to be tested: however, a less active dynein could well contribute to frequent reversals in direction (*Hancock, 2014*).

## fBm with a stochastic Hurst exponent is a new possible intracellular transport model

fBm is a Gaussian process $B_H(t)$ with zero mean and covariance $\langle B_H(t)B_H(s)\rangle \sim t^{2H} + s^{2H} - (t-s)^{2H}$, where the Hurst exponent, $H$ is a constant between 0 and 1. With the DLFNN providing local estimates of the Hurst exponent, the motion of endosomes and lysosomes can be described as fBm with a stochastic Hurst exponent, $H(t)$. This is different to multifractional Brownian motion (*Peltier and Lévy Véhel, 1995*) where $H(t)$ is a function of time. In our case, $H(t)$ is itself a stochastic process and such a process has been considered theoretically (*Ayache and Taqqu, 2005*). This is the first application of such a theory to intracellular transport and opens a new method for characterizing vesicular movement. Furthermore, *Figure 3* shows that the motion of a vesicle, $B_H(t)$, exhibits regime switching behavior between persistent and anti-persistent states.

We found that the times that lysosomes and endosomes spend in a persistent and anti-persistent state are heavy-tailed (*Figure 4*). These times are characterized by the probability densities $\psi(t) \sim t^{-\mu-1}$, where anti-persistent states have $0 < \mu < 1$ and persistent states have $1 < \mu < 2$. Extensive plots and fittings are shown in *Figure 4* and Appendix C. In fact, the residence time probability density has an infinite mean to remain in an anti-persistent state ($0<H(t)<1/2$) but in persistent states ($1/2<H(t)<1$) the mean of the residence time probability density is finite and the second moment is infinite. This implies that the vesicles may have a biological mechanism to prioritize certain interactions within the complex cytoplasm, similar to ecological searching patterns (*Reynolds and Rhodes, 2009*), mRNPs (*Song et al., 2018*), swarming bacteria (*Ariel et al., 2015*) and how human dynamics are often heavy tailed and bursty (*Barabási, 2005*).

## Conclusions

We developed a Deep Learning Feedforward Neural Network trained on fBm that estimates accurately the Hurst exponent for heterogeneous trajectories. Estimating the Hurst exponent using a DLFNN is not only more accurate than conventional methods but also enables direct trajectory

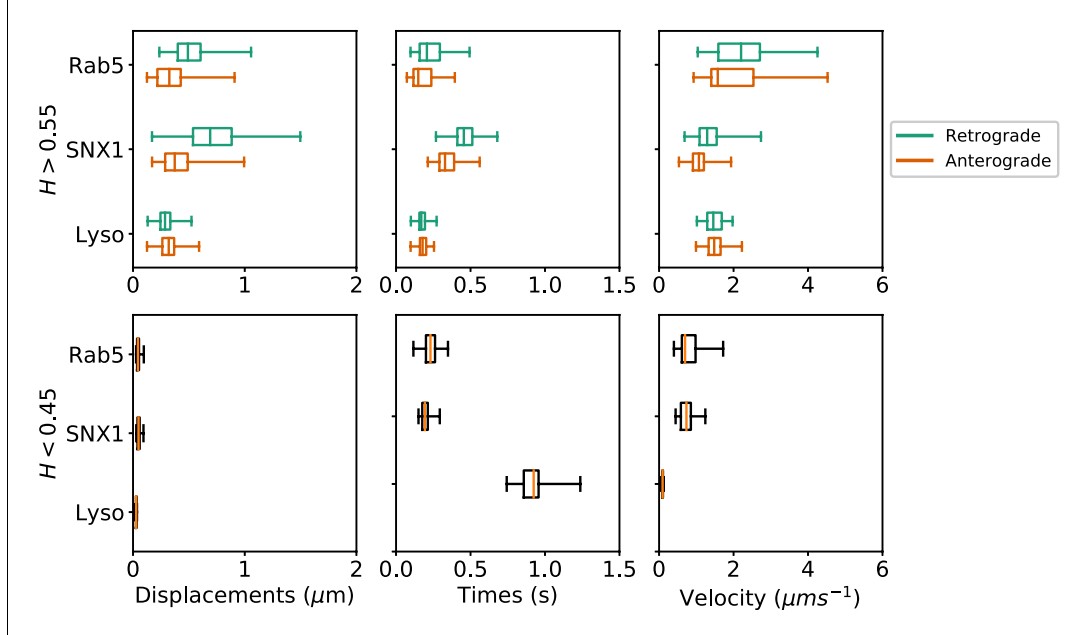

**Figure 7.** Box and whisker plots of displacements, times and velocities of persistent retrograde, persistent anterograde and anti-persistent segments in experimental trajectories. Any segment with $H>0.55$ was classed as persistent and $H<0.45$ as anti-persistent. These H values were chosen as a precaution against the mean error of the neural network estimation. Each data point within the box and whisker plots are averages of all trajectory segments in a single cell. A total of 65 MRC-5 cells for GFP-Rab5-tagged endosomes, 63 MRC-5 cells for SNX1-GFP-tagged endosomes and 71 MRC-5 cells for lysosomes were analysed with at least 5 to 500 (average 54) anterograde or retrograde segments for each cell.

segmentation without a drastic increase in computational cost. We package this DLFNN analysis code into a user-friendly application, which can predict the Hurst exponent with consistent accuracy for as few as seven consecutive data points. This is useful to biologists since major limitations to trajectory analysis are: the brevity of tracks due to the fact that particles may rapidly switch between motile states or move out of the plane of focus; the rapid nature of some biochemical reactions; and the bleaching of fluorescent probes (with non-bleaching probes often being bulky or cytotoxic). This method can be used to detect persistent and anti-persistent states of motion purely from the positional data of trajectories and removes the prerequisite of time or ensemble averaging for effective heterogeneous transport characterization.

The DLFNN enabled us to discover regime switching in lysosome and endosome movement that can be modeled by fBm with a stochastic Hurst exponent. This interpretation is a unified approach to describe motion with anti-persistence and persistence varying over time. Furthermore, the residence time of vesicles in a persistent or anti-persistent state is found to be heavy tailed, which implies that endosomes and lysosomes possess biological mechanisms to prioritize varying biological processes similar to ecological searching patterns (*Reynolds and Rhodes, 2009*), mRNPs (*Song et al., 2018*), swarming bacteria (*Ariel et al., 2015*) and even human dynamics (*Barabási, 2005*). Importantly, applying this method to identify and analyze the anterograde and retrograde motility reveals unexpected differences in behavior between closely-related organelles. Finally, in addition to providing a new segmentation method of active and passive transport, this new technique distinguishes the difference in motility between lysosomes, Rab5-positive endosomes and SNX1 positive endosomes. The results suggest that the manner in which these vesicles move is dependent on their identity within the endocytic pathway, especially when the motion is anti-persistent. This implies that directionality and the correlation between consecutive steps is important to measure in addition to the displacement, velocity and duration of movement. There is considerable scope for using these methods to identify changes in motility of different organelles caused by disease. We hope that this type of analysis will allow discoveries in particle motility of a more refined nature and make applying anomalous transport theory more accessible to researchers in a wide variety of disciplines.

# Materials and methods

### Key resources table

| Reagent type or resource | Designation | Source | Identifiers | Additional information |
|---|---|---|---|---|
| Cell line (*Homo sapiens*) | Lung fibroblast line | *Allison et al., 2017* https://doi.org/10.1083/jcb.201609033 | GFP-SNX1-MRC5 | MRC5 cell line stably expressing GFP-SNX1. Mycoplasma free. |
| Cell line (*H. sapiens*) | Lung fibroblast line | Other | GFP-Rab5-MRC5 | MRC5 cell line stably expressing GFP-Rab5 generated by retroviral transduction by G. Pearson and E. Reid, University of Cambridge. Mycoplasma free. |
| Cell line (*H. sapiens*) | MRC-5 SV1 TG1 Lung fibroblast line | ECACC | MRC-5 SV1 TG1 cells, cat no. 85042501 | Mycoplasma free. |
| Antibody | Anti-human Rab5A Rabbit monoclonal | Cell Signalling Technology | 3547S | IF(1/200) |
| Antibody | Anti-human sorting nexin 1 (mouse monoclonal) | BD Biosciences | 611482 | IF(1/200) |
| Antibody | Alexa594-conjugated anti-mouse IgG (donkey polyclonal) | Jackson ImmunoResearch | 715-585-150 | IF(1/400) |
| Antibody | A488-conjugated donkey anti-rabbit IgG | Jackson Immunoresearch | 711-545-152 | IF(1/400) |
| Recombinant DNA reagent | pLXIN-GFP-Rab5C-I-NeoR | Other | | Used by G. Pearson and E. Reid, University of Cambridge to generate retrovirus containing GFP-Rab5C |
| Sequence-based reagent | Hpa1 GFP Forward | Other | PCR primer | Used by G. Pearson and E. Reid, University of Cambridge to generate retrovirus containing GFP-Rab5C. TAGGGAGTTAACATGGTGAGCAAGGGCGAGGA |
| Sequence-based reagent | Not1 Rab5C Reverse | Other | PCR primer | Used by G. Pearson and E. Reid, University of Cambridge to generate retrovirus containing GFP-Rab5C . ATCCCTGCGGCCGCTCAGTTGCTGCAGCACTGGC |
| Chemical compound, drug | DAPI | Biolegend | 422801 | IF (1 μg/mL) |
| Chemical compound, drug | Prolong Gold | ThermoFisher | P36930 | |
| Chemical compound, drug | Lysobrite Red | AAT Bioquest | 22645 | (1/2500) |
| Chemical compound, drug | Geneticin (G418) | Sigma-Aldrich | G1397 | 200 μg/mL to maintain GFP-Rab5-MRC5 and GFP-SNX1-MRC5 cells in culture. |
| Chemical compound, drug | Formaldehyde solution, 37% (wt/v) | Sigma-Aldrich | 252549 | |
| Chemical compound, drug | Triton X-100 | Anatrace | T1001 | |
| Software, algorithm | NNT (aitracker.net) | *Newby et al., 2018* | AITracker | Web-based automated tracking service |
| Software, algorithm | Metamorph | Molecular Devices LLC | Metamorph | Metamorph Microscopy Automation and Image Analysis Software |

*Continued on next page*

*Continued*

| Reagent type or resource | Designation | Source | Identifiers | Additional information |
|---|---|---|---|---|
| Software, algorithm | FIJI | Schindelin, J.; Arganda-Carreras, I. and Frise, E. et al. (2012) , 'Fiji: an open-source platform for biological-image analysis', Nature methods 9 (7): 676–682, PMID22743772, DOI: 10.1038/nmeth.2019 | FIJI/ImageJ | |
| Software, algorithm | DLFNN Exponent Estimator | Han, Daniel. (2020, January 20). DLFNN Exponent Estimator (Version 0). http://doi.org/10.1101/777615 | DLFNN/DLFNN Exponent Estimator | Hurst exponent estimator with Deep Learning Feed-forward Neural Network application for Windows 10. Documentation included. |
| Software, algorithm | Python3 | Python Software Foundation.Python Language Reference 3.7. Available at www.python.org | Python/Python3 | |
| Software, algorithm | SciPy | Virtanen et al. (2020) SciPy 1.0: Fundamental Algorithms for Scientific Computing in Python. Nature Methods, in press. | SciPy/scipy | |
| Software, algorithm | Tensorflow | **Abadi et al., 2016** | Tensorflow | |
| Software, algorithm | Keras | Chollet, François and others. 'Keras.' (2015). Available from https://keras.io | Keras | |
| Software, algorithm | fbm | Flynn, Christopher, fbm 0.3.0 available for download at https://pypi.org/project/fbm/ or https://github.com/crflynn/fbm | FBM package in Python | Exact methods for simulating fractional Brownian motion (fBm) or fractional Gaussian noise (fGn) in python. Approximate simulation of multifractional Brownian motion (mBm) or multifractional Gaussian noise (mGn). |
| Other | 35 mm glass-bottomed dishes (μ-Dish) | Ibidi | Cat. No. 81150 | |

## Hurst exponent estimation methods

Time averaged MSDs were calculated using

$$\langle x^2(n\delta t)\rangle = \frac{1}{N-n}\sum_{m=0}^{N-n}[x((m+n)\delta t) - x(m\delta t)]^2 \tag{1}$$

where $x(n\delta t)$ is the track displacement at time $n\delta t$ and a track contains $N$ coordinates spaced at regular time intervals of $\delta t$. From now on, $\langle x\rangle$ will denote the time average of $x$ unless explicitly specified otherwise. The total time is $T=(N-1)\delta t$ and $n=1,2,...,N-1$. Lag-times are the set of possible $n\delta t$ within the data set and $\langle x^2(n\delta t)\rangle$ was then fit to a power-law $\sim t^{2H}$ using the 'scipy.optimize' package in Python3 to estimate the exponent $H$.

Rescaled ranges were calculated by creating a mean adjusted cumulative deviate series $z(n\delta t) = \sum_{m=0}^{n} x(m\delta t) - \langle x\rangle$ from original displacements $x(n\delta t)$ and mean displacement $\langle x\rangle$. Then the rescaled range is calculated by

$$[\mathrm{R/S}](n\delta t) = \frac{\max(\{z\}_n) - \min(\{z\}_n)}{\sqrt{\frac{1}{n\delta t}\sum_{m=0}^{n}(x(m\delta t) - \langle x(n\delta t)\rangle)^2}} \tag{2}$$

**Table 1.** Statistics of experimental trajectory segments.

The persistent and anti-persistent segments in this table are: from trajectories that travelled over 0.5 µm at any point from their initial starting positions; contained more points than the window size; and switched behavior more than twice in the trajectory. Note that these conditions are much stricter than those to generate *Figures 4* and *5*. Each persistent segment was then further subdivided into retrograde and anterograde segments (see Materials and methods).

| | | Rab5 | SNX1 | Lyso |
|---|---|---|---|---|
| Number of persistent segments | | 2369 | 2099 | 7645 |
| Number of anti-persistent segments | | 6983 | 3947 | 19,320 |
| Number of retrograde segments | | 2925 | 2343 | 5882 |
| Number of anterograde segments | | 2303 | 1609 | 6827 |
| Anti-persistent displacement (µm) | Mean | 0.05 | 0.05 | 0.03 |
| | Median | 0.04 | 0.05 | 0.03 |
| | St. Dev | 0.02 | 0.01 | 0.004 |
| Anti-persistent speed (µms$^{-1}$) | Mean | 0.82 | 0.75 | 0.10 |
| | Median | 0.70 | 0.73 | 0.09 |
| | St. Dev | 0.31 | 0.19 | 0.02 |
| Anti-persistent time (s) | Mean | 0.23 | 0.20 | 0.93 |
| | Median | 0.23 | 0.19 | 0.92 |
| | St. Dev | 0.05 | 0.03 | 0.11 |
| Retrograde displacement (µm) | Mean | 0.53 | 0.74 | 0.29 |
| | Median | 0.49 | 0.69 | 0.29 |
| | St. Dev | 0.19 | 0.28 | 0.08 |
| Retrograde speed (µms$^{-1}$) | Mean | 2.29 | 1.35 | 1.49 |
| | Median | 2.21 | 1.29 | 1.46 |
| | St. Dev | 0.87 | 0.39 | 0.25 |
| Retrograde time (s) | Mean | 0.22 | 0.46 | 0.17 |
| | Median | 0.21 | 0.45 | 0.17 |
| | St. Dev | 0.09 | 0.09 | 0.03 |
| Anterograde displacement (µm) | Mean | 0.35 | 0.43 | 0.31 |
| | Median | 0.33 | 0.37 | 0.32 |
| | St. Dev | 0.17 | 0.20 | 0.08 |
| Anterograde speed (µms$^{-1}$) | Mean | 2.06 | 1.10 | 1.51 |
| | Median | 1.71 | 1.08 | 1.48 |
| | St. Dev | 0.95 | 0.30 | 0.27 |
| Anterograde time (s) | Mean | 0.18 | 0.34 | 0.18 |
| | Median | 0.15 | 0.33 | 0.18 |
| | St. Dev | 0.08 | 0.08 | 0.03 |

where $\{z\}_n = z(0), z(\delta t), z(2\delta t), ..., z(n\delta t)$. The rescaled range is then fitted to a power law $[\mathrm{R/S}](n\delta t) \sim (n\delta t)^H$ where $H$ is the **Hurst (1951)**. The 'compute_Hc' function in the 'hurst' package in Python3 estimates the Hurst exponent in this way.

Sequential ranges are defined as

$$M(n\delta t) = \sup_{0 \leq s \leq n\delta t} (x(s) - x(0)) - \inf_{0 \leq s \leq n\delta t} (x(s) - x(0)) \tag{3}$$

where $\sup(x)$ is the supremum and $\inf(x)$ is the infimum for the set $x$ of real numbers. Then $M(n\delta t) = (n\delta t)^H M(\delta t)$ **Feller (1951)**.

## DLFNN structure and training

The fractional Brownian trajectories were generated using the Hosking method within the 'FBM' function available from the 'fbm' package in Python3. The DLFNN was built using Tensorflow *Abadi et al. (2016)* and Keras *Chollet (2015)* in Python3 and trained by using the simulated fractional Brownian trajectories. The training and testing of the neural network were performed on a workstation PC equipped with 2 CPUs with 32 cores (Intel(R) Xeon CPU E5-2640 v3) and 1 GPU (NVIDIA Tesla V100 with 16 GB memory). The structure of the neural network was a multilayer, feedforward neural network where all nodes of the previous layer were densely connected to nodes of the next layer. Each node had a ReLU activation function and the parameters were optimized using the RMSprop optimizer (see Keras documentation *Chollet, 2015*). Three separate structures were explored and examples of these structures for two hidden layers and five time point inputs are shown in *Figure 1g,h and i*. The triangular structure was predominantly used since this was the least computationally expensive and accuracy between different structures were similar. To compare the accuracy of different methods, the mean absolute error ($\sigma_H$) of $N$ trajectories, $\sigma_H = \sum_{m=1}^{N} \left( H_n^{sim} - H_n^{est} \right)/N$, was used. Before inputting values into the neural network, the time series was differenced to make it stationary. The input values of a fBm trajectory $\{x\} = x_0, x_1, ..., x_n$ were differenced and normalized so that $\{x_{input}\} = (x_1 - x_0)/\text{range}(x), (x_2 - x_1)/\text{range}(x), ..., (x_n - x_{n-1})/\text{range}(x)$. Since the model requires differenced and normalized input values, in theory it should be applicable to a wide range of datasets. However, further testing must be done in order to confirm this expectation.

## Gaussian kernel density estimation

Kernel density estimation (KDE) is a non-parametric method to estimate the probability density function (PDF) of random variables. If $N$ random variables $x_n$ are distributed by an unknown density function $P(x)$, then the kernel density estimate $P(x)$ is

$$\hat{P}(x) = \frac{1}{N} \sum_{n=1}^{N} K\left(\frac{x - x_n}{l}\right) \tag{4}$$

where $K(\cdot)$ is the kernel function and $l$ is the bandwidth. In this paper, we have used a Gaussian KDE, $K(y) = \frac{1}{\sqrt{2\pi}} e^{-y^2/2}$, to estimate the two dimensional PDFs of the second and bottom row in *Figure 1a*. This was performed in Python3 using 'scipy.stats.gaussian_kde' and Scott's rule of thumb for bandwidth selection.

## Segmenting trajectories into persistent and anti-persistent segments

From the estimates of Hurst exponent from the DLFNN, trajectories were segmented into persistent and anti-persistent segments. Given an experimental trajectory $x = x_0, x_1, ..., x_n$ and window of length $N_w$ (an odd number) starting at $x_i$, we obtain the $H$ estimate for the position at $x_j$, where $j = i + (N_w - 1)/2$. This will give us a series of $H_t$ values, $H_{(N_w-1)/2}, H_{(N_w-1)/2+1}, ..., H_{n-(N_w-1)/2}$, which correspond to the positions, $x_{(N_w-1)/2}, x_{(N_w-1)/2+1}, ..., x_{n-(N_w-1)/2}$. Then, the values $H_t$ can be segmented into consecutive points of persistence $H_t > 0.55$ and anti-persistence $H_t < 0.45$. The bounding values, 0.55 and 0.45, were used since the mean error of the DLFNN estimation method was $\sigma_H \sim 0.05$. Any segment less than the length of $N_w$ was discarded as a precaution against spurious detection.

## Directional segmentation of persistent segments

Once segments of persistence and anti-persistence were defined, we measured the displacement, time and velocity of these segments, shown in the bottom row of *Figure 7* and *Table 1*. The persistent segments were filtered to be only from trajectories that travelled over 0.5 μm; contained more points than the window size; and switched behaviour more than twice in the trajectory. In addition, we assessed if persistent segments were anterograde or retrograde in direction. In order to do this, the centrosomal region was defined by the user as the point where the lysosomes, Rab5 and SNX1 organelles were the largest, brightest, or the most clustered. Image contrast enhancements, such as histogram equalization, were used to locate the centrosomes. By locating the centrosomal region and the cell boundary from user input, the persistent segments can then be classified as anterograde or retrograde. This was done by finding the cosine of the angles, $\cos(\theta)$, between the vector, $\vec{r}_{0,i}$,

from the centrosome to the current particle position $x_i$ and the vector, $\vec{r}_{i,i+1}$, from the current particle position to the next particle position $x_{i+1}$. The exact formula is $\cos(\theta) = \vec{r}_{0,i} \cdot \vec{r}_{i,i+1}/|\vec{r}_{0,i}||\vec{r}_{i,i+1}|$. Using windows in a similar fashion as determining persistent and anti-persistent segments, $\cos(\theta_i)$ corresponding to position $x_i$ was found for the points within a persistent segment. If $\cos(\theta_i) > \sigma_{\cos(\theta)}$, then the motion was deemed to be anterograde and if $\cos(\theta_i) < -\sigma_{\cos(\theta)}$, retrograde. Sweeping through the points of $x_i$, consecutive retrograde or anterograde points formed segments from the persistent segments. A threshold of $\sigma_{\cos(\theta)} = 0.3$ was used.

## Cell lines

The MRC-5 SV1 TG1 Lung fibroblast cell line was purchased from ECACC. MRC-5 cell lines stably expressing GFP-Rab5C and GFP-SNX1 were kindly provided by Drs. Guy Pearson and Evan Reid (Cambridge Institute for Medical Research, University of Cambridge). The GFP-SNX1 cell line has been previously described in *Allison et al. (2017)*. Cell lines were routinely tested for mycoplasma infection. To generate the MRC-5 GFP-Rab5C stable cell line, GFP-Rab5C was PCRed from pIRES GFP-Rab5C *Seaman (2004)* using 'Hpa1 GFP Forward' (TAGGGAGTTAACATGGTGAGCAAGGGC-GAGGA) and 'Not1 Rab5C Reverse' (ATCCCTGCGGCCGCTCAGTTGCTGCAGCACTGGC) oligonucleotide primers. The GFP-Rab5C PCR product and a pLXIN-I-NeoR plasmid were digested using Hpa1 (New England Biolabs - R0105) and Not1 (New England Biolabs - R3189) restriction enzymes. The GFP-Rab5C PCR product was then ligated into the digested pLXIN-I-NeoR using T4 DNA Ligase (New England Biolabs - M0202). The ligated plasmid was amplified in bacteria selected with ampicillin and verified using Sanger Sequencing. To generate the GFP-Rab5C MRC-5 cell line, Phoenix retrovirus producer HEK293T cells were transfected with the pLXIN-GFP-Rab5C-I-NeoR plasmid to generate retrovirus containing GFP-Rab5C. MRC-5 cells were inoculated with the virus, and successfully transduced cells were selected using 200 µg/mL Geneticin (G418 - Sigma-Aldrich G1397). Cells used for imaging were not clonally selected.

## Live-imaging and tracking

Stably expressing MRC-5 cells were co-stained with LysoBrite Red (AAT Bioquest), imaged live using fluorescence microscopy and tracked with NNT aitracker.net; *Newby et al. (2018)*. The cells were grown in MEM (Sigma Life Science) and 10% FBS (HyClone) and incubated for 48 hr at 37 in 5% $CO_2$ on 35 mm glass-bottomed dishes (µ-Dish, Ibidi, Cat. No. 81150). For LysoBrite staining, LysoBrite was diluted 1 in 500 with Hank's Balanced Salt solution (Sigma Life Science). Then 0.5 mL of this solution was added to cells on a 35 mm dish containing 2 mL of growing media and incubated at 37 for at least 1 hr. Cells were then washed with sterile PBS and the media replaced with growing media.

After at least 6 hr incubation, the growing media was replaced with live-imaging media composed of Hank's Balanced Salt solution (Sigma Life Science, Cat. No. H8264) with added essential and non-essential amino acids, glutamine, penicillin/streptomycin, 25 mM HEPES (pH 7.0) and 10% FBS (HyClone). Live-cell imaging was performed on an inverted Olympus IX71 microscope with an Olympus 100 × 1.35 oil PH3 objective. Samples were illuminated using an OptoLED (Cairn Research) light source with 470 nm and white LEDs. For GFP, a 470 nm LED and Chroma ET470/40 excitation filter was used in combination with a Semrock FITC-3540C filter set. For Lysobrite-Red, a white light LED, Chroma ET573/35 was used with a dualband GFP/mCherry dichroic and an mCherry emission filter (ET632/60). GFP-Rab5-labeled endosomes were imaged in a total of 65 cells, from three independent experiments. GFP-SNX1-labeled endosomes were imaged in a total of 63 cells from four independent experiments. Lysosomes were imaged in separate experiments, with 71 cells imaged from three independent repeats. A stream of 20 ms exposures was collected with a Prime 95B sCMOS Camera (Photometrics) for 17 s using Metamorph software while the cells were kept at 37 (in atmospheric CO2 levels). The endosomes and lysosomes in the videos were then tracked using an automated tracking software (AITracker) *Newby et al. (2018)*.

## Confocal imaging

To compare the localization of SNX1 and Rab5, GFP-Rab5-MRC-5 cells were grown on #1.5 coverslips and then fixed in 3% (w/v) formaldehyde in PBS for 20 min at room temperature (RT). Coverslips were washed twice in PBS, quenched in PBS with glycine, then permeabilized by incubation for 5 min in 0.1% Triton X-100. After another wash in PBS, coverslips were labeled with antibodies to

SNX1 and Rab5 for 1 h at RT, washed three times in PBS, then labeled with Alexa488-donkey anti-rabbit and Alexa594-donkey anti-mouse antibodies in 1 μg/mL DAPI in PBS for 30 min. After three PBS washes, coverslips were dipped in deionized water, air-dried and mounted on slides using Prolong Gold.

Images were collected on a Leica TCS SP8 AOBS inverted confocal using a 100x/1.40 NA PL apo objective. The confocal settings were as follows: pinhole, one airy unit; scan speed 400 Hz unidirectional; format 2048 × 2048. Images were collected using hybrid detectors (A488 and A594) or a PMT (DAPI) with these detection mirror settings; [Alexa488, 498 nm-577 nm; Alexa594, 602 nm-667 nm; DAPI, 420 nm-466 nm] using the SuperK Extreme supercontinuum white light laser for 488 nm (10.5%) and 594 nm (5%) excitation, and a 405 nm laser (5%) for DAPI. Images were collected sequentially to eliminate cross-talk between channels. When acquiring 3D optical stacks the confocal software was used to determine the optimal number of Z sections. The data were deconvolved using Huygens software before generating maximum intensity projections of 3D stacks using FIJI.

## Software and code

The code and documentation for determining the Hurst exponent can be found in https://github.com/dadanhan/hurst-exp (copy archived at https://github.com/elifesciences-publications/hurst-exp; *Han, 2019*) and a GUI is available on https://zenodo.org/record/3613843#.XkPf2Wj7SUl.

## Acknowledgements

The authors thank: Dr. Jay Newby and Zach Richardson (UNC Chapel Hill) for assistance in using the automated tracking system (AITracker); Prof. Philip Woodman, Prof. Martin Lowe, Prof. Matthias Weiss (Universität Bayreuth), Dr. Henry Cox, Dr. Jack Hart, Rebecca Yarwood and Hannah Perkins for discussions; and, Dr. Guy Pearson and Dr. Evan Reid (University of Cambridge) for providing the GFP-Rab5 and GFP-SNX1 MRC-5 cell lines. DH acknowledges financial support from the Wellcome Trust Grant No. 215189/Z/19/Z. SF, NK, TAW, and VJA acknowledge financial support from EPSRC Grant No. EP/J019526/1. VJA acknowledges support from the Biotechnology and Biological Sciences Research Council grant number BB/H017828/1. AG acknowledges financial support from the Wellcome Trust Grant No. 108867/Z/15/Z. The Leica SP8 microscope used in this study was purchased by the University of Manchester Strategic Fund. Special thanks goes to Dr. Peter March for his help with the confocal imaging.

## Additional information

### Funding

| Funder | Grant reference number | Author |
| --- | --- | --- |
| Wellcome Trust | 215189/Z/19/Z | Daniel Han |
| EPSRC | EP/J019526/1 | Nickolay Korabel<br>Victoria J Allan<br>Sergei Fedotov<br>Thomas A Waigh |
| BBSRC | BB/H017828/1 | Victoria J Allan |
| Wellcome Trust | 108867/Z/15/Z | Anna Gavrilova |

The funders had no role in study design, data collection and interpretation, or the decision to submit the work for publication.

### Author contributions

Daniel Han, Conceptualization, Data curation, Software, Formal analysis, Validation, Investigation, Visualization, Methodology, Writing - original draft, Project administration, Writing - review and editing; Nickolay Korabel, Formal analysis, Validation, Investigation, Writing - review and editing; Runze Chen, Software, Formal analysis; Mark Johnston, Validation, Investigation; Anna Gavrilova, Data

curation, Investigation; Victoria J Allan, Conceptualization, Resources, Data curation, Formal analysis, Supervision, Funding acquisition, Validation, Investigation, Visualization, Methodology, Project administration, Writing - review and editing; Sergei Fedotov, Conceptualization, Resources, Formal analysis, Supervision, Funding acquisition, Validation, Methodology, Project administration, Writing - review and editing; Thomas A Waigh, Conceptualization, Supervision, Validation, Methodology, Project administration, Writing - review and editing

## Author ORCIDs
Daniel Han (iD) https://orcid.org/0000-0002-9088-1651
Victoria J Allan (iD) https://orcid.org/0000-0003-4583-0836
Thomas A Waigh (iD) https://orcid.org/0000-0002-7084-559X

## Decision letter and Author response
Decision letter https://doi.org/10.7554/eLife.52224.sa1
Author response https://doi.org/10.7554/eLife.52224.sa2

# Additional files
## Supplementary files
• Transparent reporting form

## Data availability
Supporting files are on GitHub and Zenodo.

The following dataset was generated:

| Author(s) | Year | Dataset title | Dataset URL | Database and Identifier |
|---|---|---|---|---|
| Daniel Han | 2020 | DLFNN Exponent Estimator | https://zenodo.org/record/3613843#.XmzM0pP7RRY | Zenodo, 3613843#.XmzM0pP7RRY |

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

**Appendix 1**

## Comparison of HMM model against fBm model segmentation of experimental trajectories

In order to compare the effectiveness of the neural network and hidden Markov models (HMM), qualitative plots were made of real trajectories and their respective comparisions. The models in the HMM analysis approach had a maximum of three different motion states. It is clear from comparing the endosome track segmented using DLFNN (*Figure 3*) with *Appendix 1—figures 1–3*, that segmentation using hidden Markov models is not suitable for endosome trajectories. Perhaps, by increasing the number of states within models, the hidden Markov models can achieve similar results of the neural network, but this analysis becomes computationally expensive.

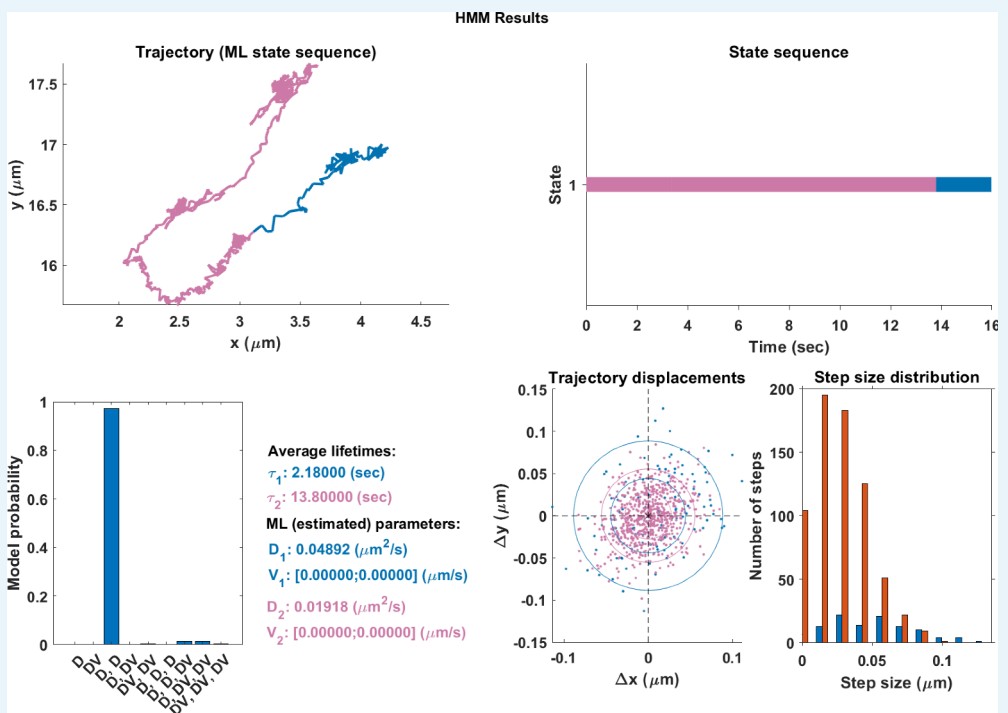

**Appendix 1—figure 1.** The same single trajectory as in *Figure 3* but processed using the HMM-Bayes package described in *Monnier et al. (2015)*. The plot shows: the original trajectory (*top left*); the inferred state sequence of the most likely model (*top right*); the model probabilities given a maximum of three possible states (*bottom left*); the average lifetime of each state and estimated parameters of the most likely model (*bottom center left*), which in this case is two different diffusive states; individual increment displacements (*bottom center right*); and the step size distribution of those increments classed into the two different states.

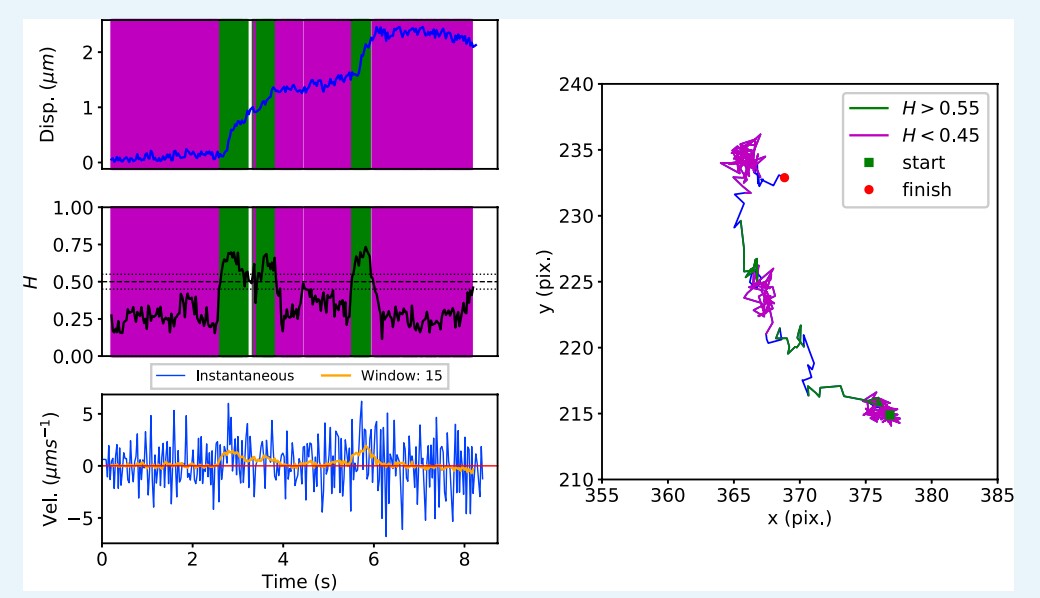

**Appendix 1—figure 2.** *Top*: Plot of displacement from a single GFP-SNX1 endosome trajectory in an MRC-5 cell (blue). Shaded areas show persistent ($0.55 < H < 1$ in green) and anti-persistent ($0 < H < 0.45$ in magenta) behaviour. *Middle*: A 15 point moving window DLFNN exponent estimate for the trajectory (black) with a line (dashed) marking diffusion $H = 0.5$ and lines (dotted) marking the confidence bounds $H = 0.55$ and $0.45$. *Bottom*: Plot of instantaneous and moving (15 point) window velocity. *Right*: Plot of the trajectory of a GFP-SNX1 endosome in an MRC-5 cell with start and finish positions, and persistent (green) and anti-persistent (magenta) segments indicated.

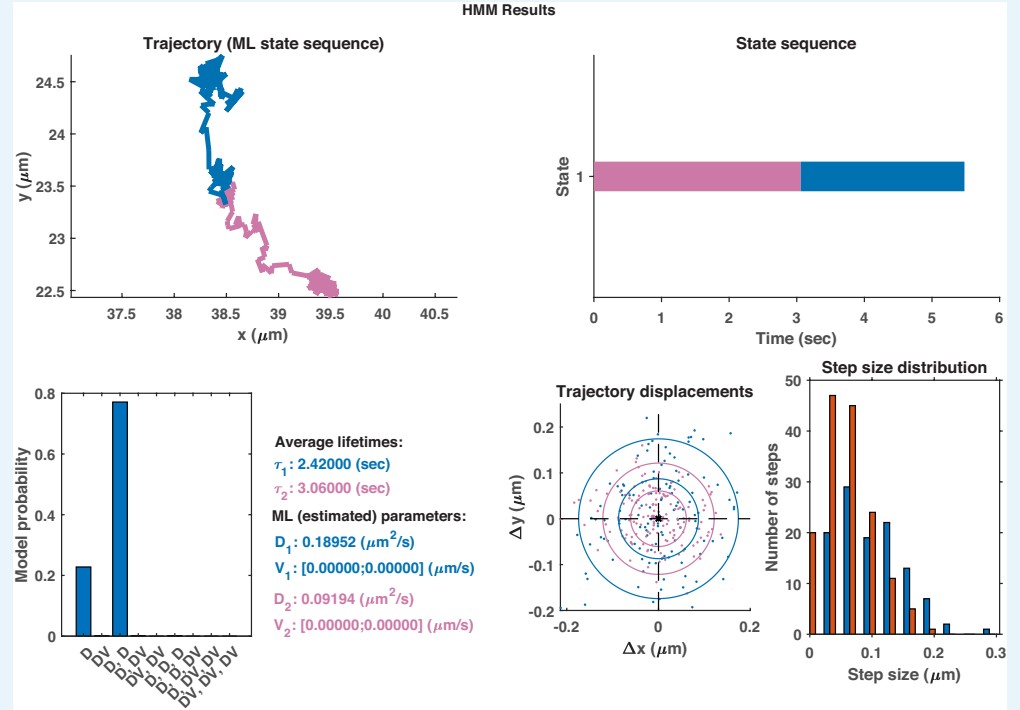

**Appendix 1—figure 3.** The same HMM-Bayes analysis as shown in *Appendix 1—figure 1* applied to the trajectory in *Appendix 1—figure 2*.

## Appendix 2

### Testing DLFNN accuracy for different diffusion coefficients

The DLFNN was compared to the MSD estimation method for simulated trajectories with different diffusion coefficients to ensure that the DLFNN estimation was not scale dependent.

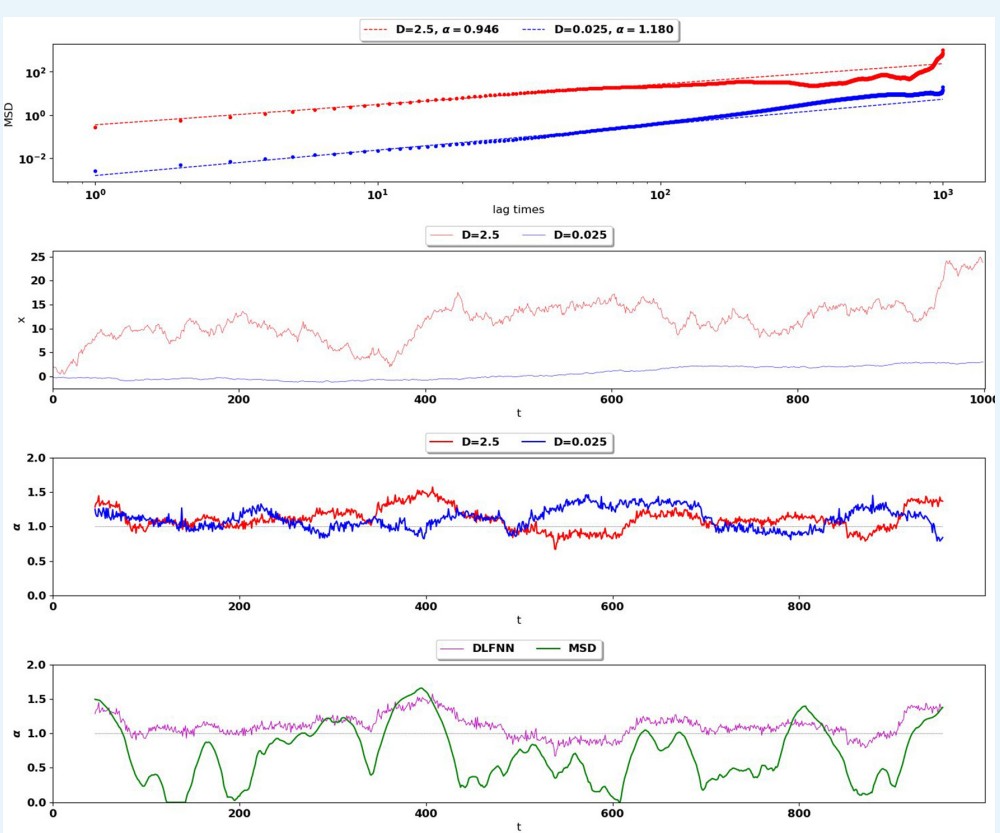

**Appendix 2—figure 1.** Top: MSD (points) and power-law fits (dashed) for two different Brownian trajectories containing 1000 data points with diffusion coefficient 2.5 (red) and 0.025 (blue). The Hurst exponent should be $\alpha = 2H = 1$ for both trajectories. Second Row: Simulation of the two Brownian trajectories with diffusion co-efficient 2.5 (red) and 0.025 (blue). Third Row: Local Hurst exponent estimates given by the DLFNN for the two different trajectories using a 90 point window. The averages of DLFNN Hurst exponent estimates are $\alpha = 2H = 1.110$ (red) and $\alpha = 2H = 1.114$ (blue). Bottom: Local Hurst exponent estimates of the $D = 2.5$ track given by DLFNN and MSDs using a 90 point window. The average of DLFNN Hurst exponent estimates is $\alpha = 2H = 1.110$ and the average of MSD estimates is $\alpha = 2H = 0.937$.

## Appendix 3

### Measuring the residence time and flight length probability density functions of persistent and anti-persistent states

Classifying persistent and anti-persistent states by Hurst exponent values $1/2 < H < 1$ and $0 < H < 1/2$ respectively, individual lysosome and endosome trajectories were segmented with a moving window of 15 points. Data was extracted from microscopy movies from three independent experiments. GFP-Rab5 endosomes from 65 cells, GFP-SNX1 endosomes from 63 cells and lysosomes from 71 different cells were tracked using AITracker *Newby et al. (2018)*. Then the trajectories were segmented into anti-persistent ($0 < H < 1/2$) and persistent ($1/2 < H < 1$) using the Hurst exponent estimates by DLFNN. The time duration and particle displacement of these segments were measured and then fitted to distributions. In this way, we could measure the stochastic switching between active and passive transport and the statistics of vesicle movement within these states.

*Figure 4* shows the survival time probabilities $\Psi(t)$ of different states of motion (persistent and anti-persistent) in vesicle trajectories. $\Psi(t)$ is the probability that the vesicle will still be in the same state of motion after time $t$ has elapsed. *Figure 4* shows that the persistent and anti-persistent states follow $\Psi(t) = e^{-\lambda t} \left( \frac{\tau_0}{\tau_0 + t} \right)^{\mu}$. The survival time probabilities show that lysosomes and endosomes are far more likely to remain trapped in a anti-persistent state than be persistently transported by motor proteins. While this is intuitively obvious in the context of cell biology, this analysis provides quantitative characterization of endosomal and lysosomal motility. *Appendix 3—table 1* shows the parameters of fitting for *Figure 4*. *Appendix 3—figure 1* shows the empirical probability density functions (PDF) of the particle displacements for different states of motion. Displacements of segments are fitted to Burr Type XII distributions,

$$P(x) = \frac{kc(x/x_0)^{c-1}}{x_0 \left( 1 + (x/x_0)^c \right)^{k+1}}$$

where $x$, $x_0$, $c$ and $k > 0$.

As expected, this analysis demonstrates that both endosomes and lysosomes are far more likely to move large distances when they are in the persistent state. This reconciles how vesicles are able to move large distances even though they are more likely to stay in a anti-persistent state for long periods of time. The large displacements in the persistent state compete with long durations spent in the anti-persistent state.

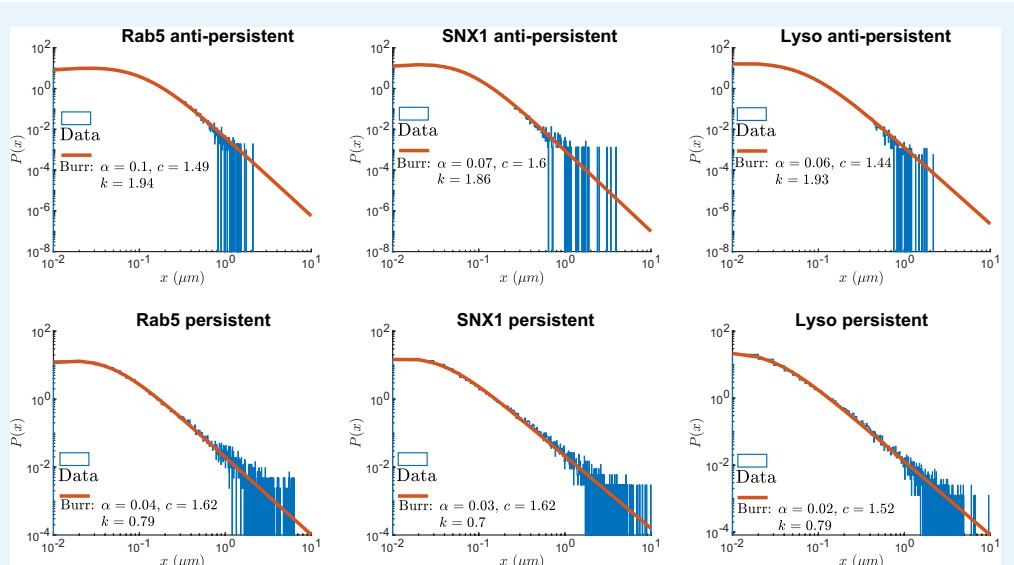

**Appendix 3—figure 1.** Normalized histograms (blue) and corresponding maximum likelihood estimation for Burr distributions (line) of segment displacements from lysosome and endosome experimental trajectories segmented using DLFNN. Parameter estimates are shown the legend.

**Appendix 3—table 1.** Results for the fits of survival time probabilities shown in *Figure 4*. The parameters and the analytical survival functions used to fit the Kaplan-Meier estimator survival curves.

| Rab5 data | Survival function $\Psi(t)$ | Fit parameters for *Figure 4* |
|---|---|---|
| Anti-persistent | $e^{-\lambda t}\left(\frac{\tau_0}{\tau_0+t}\right)^{\mu}$ | $\mu = 0.518 \pm 0.004,\ \tau_0 = 0.140 \pm 0.002s,\ \lambda = 0.352 \pm 0.002s^{-1}$ |
| Persistent | $e^{-\lambda t}\left(\frac{\tau_0}{\tau_0+t}\right)^{\mu}$ | $\mu = 1.352 \pm 0.102,\ \tau_0 = 0.045 \pm 0.006s,\ \lambda = 1.286 \pm 0.142s^{-1}$ |
| SNX1 data | Survival function $\Psi(t)$ | Fit parameters for *Figure 4* |
| Anti-persistent | $e^{-\lambda t}\left(\frac{\tau_0}{\tau_0+t}\right)^{\mu}$ | $\mu = 0.757 \pm 0.023,\ \tau_0 = 0.118 \pm 0.006s,\ \lambda = 1.004 \pm 0.016s^{-1}$ |
| Persistent | $e^{-\lambda t}\left(\frac{\tau_0}{\tau_0+t}\right)^{\mu}$ | $\mu = 2.034 \pm 0.205,\ \tau_0 = 0.185 \pm 0.026s,\ \lambda = 0.659 \pm 0.137s^{-1}$ |
| Lysosome data | Survival function $\Psi(t)$ | Fit parameters for *Figure 4* |
| Anti-persistent | $e^{-\lambda t}\left(\frac{\tau_0}{\tau_0+t}\right)^{\mu}$ | $\mu = 1.113 \pm 0.009,\ \tau_0 = 0.208 \pm 0.003s,$ <br> $\lambda = 0.5s^{-1}$ (fixed) |
| Persistent | $e^{-\lambda t}\left(\frac{\tau_0}{\tau_0+t}\right)^{\mu}$ | $\mu = 1.748 \pm 0.065,\ \tau_0 = 0.041 \pm 0.003s,\ \lambda = 1.216 \pm 0.139s^{-1}$ |

## Appendix 4

### Calculating information criteria for GMM fittings

In order to determine the minimum amount of components necessary to model the histograms of Hurst exponent in *Figure 5*, the Akaike and Bayes Information Criterion were computed.

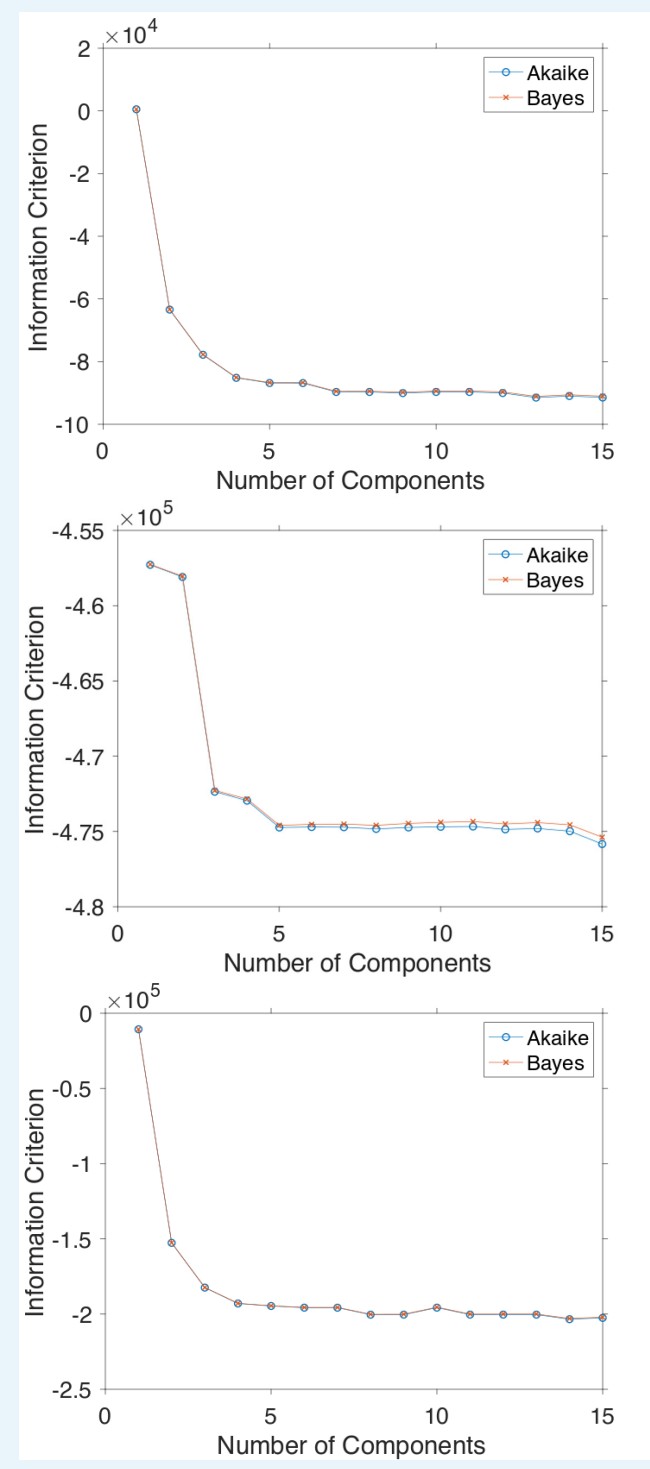

**Appendix 4—figure 1.** The Akaike and Bayes information criterion against number of components in the Gaussian mixture model shown in *Figure 5* for GFP-Rab5 tagged endosomes (top), SNX1-GFP tagged endosomes (middle) and lysosomes (bottom).

