## [Decision Letter]

**Acceptance summary:**

The authors develop a useful Deep Learning Neural Network to classify heterogeneous single-particle motion from live-cell imaging data. They apply their model to infer fractional Brownian motion from both simulated and cellular data, and show that it out-performs competing approaches such as MSD-based averaging and exponent inference, accurately predicting Hurst exponents in as few as 7 steps.

**Decision letter after peer review:**

Thank you for submitting your article "Deciphering anomalous heterogeneous intracellular transport with neural networks" for consideration by *eLife*. Your article has been reviewed by three peer reviewers, one of whom is a member of our Board of Reviewing Editors, and the evaluation has been overseen by Aleksandra Walczak as the Senior Editor. The following individuals involved in review of your submission have agreed to reveal their identity: Hyeyoon Park (Reviewer #3).

The reviewers have discussed the reviews with one another and the Reviewing Editor has drafted this decision to help you prepare a revised submission.

The authors develop a useful Deep Learning Neural Network to classify heterogeneous single-particle motion from live-cell imaging data. They apply their model to infer fractional Brownian motion from both simulated and cellular data, and show that it out-performs competing approaches such as MSD-based averaging and exponent inference, accurately predicting Hurst exponents in as few as 7 steps. Their article is well written and presented.

Summary:

All the reviewers see the merit in the approach, although there is some concern that it is not particularly novel and the approach may be biased.

Essential revisions:

The authors should either explain or illustrate how training their model solely on the class of models they seek to infer, rather than a broader set of models, may bias their inference procedure. For example, could they perform a test of this potential bias directly by training the neural network by including other models that are not fBm, and show how this impacts their results, particularly in the case of cellular datasets?

Limited comparison would be helpful for the cellular datasets in particular in order to illustrate to the reader how/why application of the fBm to infer single-particle trajectory data from living systems is useful for inferring molecular/other mechanism of dynamical molecular motion in living systems.

It would be helpful if the authors could elaborate a bit further on how their fBm and Hurst exponents are useful to infer motion mechanism.

The authors need to offer a bit more biological/cellular insight into their preceding findings on vesicular motion?

*Reviewer #1:*

The authors provide an algorithmic approach for evaluating cellular transport distinct from MSD based approaches. The useful feature is that presumably fewer points are needed for predictive value. This is important since photobleaching is a major limitation in particle tracking, the more information that can be extracted from limited data, the better.

I feel this approach may be useful, but it is not clear how this compares to the current standard HMM analysis in a head to head matchup. Also, it seems a bit cumbersome so anything they can do to make it user friendly would be appreciated.

*Reviewer #2:*

The authors develop a useful Deep Learning Neural Network to classify heterogeneous single-particle motion from live-cell imaging data. They apply their model to infer fractional Brownian motion from both simulated and cellular data, and show that it out-performs competing approaches such as MSD-based averaging and exponent inference, accurately predicting Hurst exponents in as few as 7 steps. Their article is well written and presented.

Training the neural networks on a fractional Brownian motion (fBm) model alone must effectively "bias" the inference procedure towards this model, acting analogous to a prior in Bayesian inference. Can the authors either explain or illustrate how training their model solely on the class of models they seek to infer, rather than a broader set of models, may bias their inference procedure? For example, could they perform a test of this potential bias directly by training the neural network by including other models that are not fBm, and show how this impacts their results, particularly in the case of cellular datasets?

In this vein, the authors cite HMM-based models that infer diffusion and directed motion from single-particle trajectories, but they do not compare their procedure with these methods. Some limited comparison would be helpful for the cellular datasets in particular in order to illustrate to the reader how/why application of the fBm to infer single-particle trajectory data from living systems is useful for inferring molecular/other mechanism of dynamical molecular motion in living systems.

Related to this, it would also be helpful if the authors could elaborate a bit further on how their fBm and Hurst exponents are useful to infer motion mechanism, immediately before Conclusions, where the authors write:

"This implies that the vesicles may have a biological mechanism to prioritise certain interactions within the complex cytoplasm, similar to how human dynamics are often heavy tailed and bursty Barabasi, (2005)."

Given the rather large difference between human dynamics and molecular, organelle, or vesicular dynamics, could the authors offer a bit more biological/cellular insight into their preceding findings on vesicular motion?

For a wide application of this analysis tool in biology, can the authors provide a directly executable GUI software?

*Reviewer #3:*

This paper describes an analysis tool for particle trajectory data. The authors used a deep learning feedforward neural network (DLFNN) to extract a stochastic Hurst exponent H(t) from a trajectory. They found that the neural network is a more sensitive method to characterize fractional Brownian motion (fBM) than previous statistical tools such as mean squared displacement (MSD), rescaled range, and sequential range methods. They applied this tool to analyze the trajectories of lysosome and endosomes in live cells. The topic is interesting, but the novelty and impact of the work are not very clear. Also, the software is based on Python, which may limit the application of the tool by a wide range of researchers in cell biology. A user-friendly, standalone software would be more helpful.

1) As the authors mentioned in the Introduction, exponent estimation using neural networks has been already demonstrated. The authors claimed a novelty in that the local H(t) is used to segment single trajectories into persistent and anti-persistent sections. However, the manuscript is lacking comparison with the existing methods using hidden Markov models and rolling windowed analysis. The authors also claimed that "fBM with a stochastic Hurst exponent is a new intracellular transport model". However, this section is rather brief, and some notations are not clearly defined. I think stronger impact and novelty are required for publication in *eLife*.

2) For a wide application of this analysis tool in biology, can the authors provide a directly executable GUI software?

3) It is unclear whether users can apply the pre-trained model for a broad range of data with different time and spatial scales. Or do users have to train the neural network for their own data set? More detailed instruction for the training is required. In the subsection “DLFNN structure and training”, the last sentence needs more explanation.

4) In Figure 1E, it is counterintuitive that the error (σ_H_) increases as the SNR increases. In subsection “The DLFNN is more accurate than established methods”, 'Gaussian noise with increasing signal-to-noise ratio" needs to be revised. Please use a different term for this parameter or revise the figure with a commonly used definition of SNR.

5) It would be informative if the authors also show the Hurst exponent estimated from the TAMSD method in Figure 2.

6) Heavy-tailed distributions have been observed not only in human dynamics but also in many biological systems. Some of the relevant review and original articles are listed below.

-Reynolds and Rhodes, (2009)

-Ariel, et al., (2015)

-Song et al., (2018)

-Chen et al., (2015)

---

## [Author Response]

The authors develop a useful Deep Learning Neural Network to classify heterogeneous single-particle motion from live-cell imaging data. They apply their model to infer fractional Brownian motion from both simulated and cellular data, and show that it out-performs competing approaches such as MSD-based averaging and exponent inference, accurately predicting Hurst exponents in as few as 7 steps. Their article is well written and presented.Summary:All the reviewers see the merit in the approach, although there is some concern that it is not particularly novel and the approach may be biased.

We acknowledge the concern of reviewers of novelty and potential bias in our method. However, we believe that our specifically trained neural network allows us to illuminate the heterogeneous behavior in single experimental trajectories with great accuracy. To our knowledge, this has not been achieved before by other methods. Please see our responses to the essential revisions for further details.

Essential revisions:1) The authors should either explain or illustrate how training their model solely on the class of models they seek to infer, rather than a broader set of models, may bias their inference procedure. For example, could they perform a test of this potential bias directly by training the neural network by including other models that are not fBm, and show how this impacts their results, particularly in the case of cellular datasets?

We fully agree that a broad set of possible anomalous transport models pose a challenge in selecting the optimal one, especially in deducing the origin of anomalous behaviour. However, we believe that training the neural network solely on fractional Brownian motion (fBm) is well justified for cellular datasets and especially for the short time scales in which we are seeking to classify motion. On the trajectory level, fBm behaves the same as the generalized Langevin equation (GLE) model which is a good description of equilibrium and non-equilibrium behavior of a tracer particle undergoing anomalous diffusion. This supports our choice of fBm model. Other anomalous transport models, such as scaled Brownian motion (sBm), subdiffusive continuous time random walks (CTRW) and superdiffusive L´evy walks (LW), have been shown to be good models for describing transport of intracellular vesicles on a mesoscopic scale. However, all of these models are not suitable for direct segmentation and interpretation from positions of tracked particles in consecutive frames of experimental videos on the scales smaller than single unidirectional runs.

On a microscopic level, sBm trajectories display normal Gaussian diffusion, CTRW trajectories are stationary and LW trajectories are ballistic. If we consider CTRW models, the microscopic dynamics at scales smaller than the waiting times would show the particle mostly stationary with instantaneous jumps to different locations, which is clearly unrealistic. For L´evy walk models on microscopic scales, the particle would be ballistic for the majority of times except at the turning points in the trajectory. The ‘anomalous diffusiveness’ of such random walks are thus generated in the mesoscopic scale when considering the statistics of many jumps or runs. On the other hand, fBm has consistent statistical interpretations for all scales due to its self-similar properties. Furthermore, the effectiveness of the neural network to estimate the Hurst exponent is due to the correlation of stationary increments and self-similarity. For other models based on the statistics of multiple jumps, the neural network would have similar efficiency of estimation as the standard statistical methods such as mean-square displacements (MSDs).

Experimentally, there is a strong motivation for considering transport inside cells as fBm [1, 2]. Therefore, we find that fBm (or GLE) is the model that is most physical. Furthermore, fBm can model ‘both sub-diffusion (0 *< H <* 1*/*2) and super-diffusion (1*/*2 *< H <* 1).… in a unified manner using only the Hurst exponent’ (Introduction).

In order to better address these concerns, we add the following to the Introduction:

‘fBm was chosen due to its self-similar properties that allow direct analysis at short time scales given by experimental systems; and the experimental evidence for fBm in the crowded cytoplasm [3, 1, 2]. Moreover, other anomalous diffusion models, such as scaled Brownian motion [4], subdiffusive continuous time random walks [5] and superdiffusive L´evy walks [6] are not suitable to interpret anomalous trajectories on the microscopic level.’

2) Limited comparison would be helpful for the cellular datasets in particular in order to illustrate to the reader how/why application of the fBm to infer single-particle trajectory data from living systems is useful for inferring molecular/other mechanism of dynamical molecular motion in living systems.

There are several other models of inferring vesicle motion in the cytoplasm, most notably the HMM-Bayes model introduced by Monnier et al., 2017. To address the concern about the usefulness of our method, we compare results from HMM-Bayes package to our results. The HMM-Bayes model was applied to the same trajectory as shown in Figure 3. The result in the newly added Appendix 1 clearly shows that the HMM-Bayes model is less suitable for segmenting anomalous intracellular trajectories of endosomes and lysosomes since the active motility was not picked up properly. It is apparent that the fBm description with persistent and anti-persistent movement produces more intuitive results than that of HMM Bayes with multiple velocity and diffusive states. Quantitatively, our results for subdiffusion and superdiffusion segmentation show residence time probability density functions that are consistent with the truncated power-law distributed run times found in [Chen, Wang and Granick, 2015] (Figure 4) albeit they use a directionally persistent ecological method to detect single runs.

The comparisons are made in Appendix 1, a newly added section, and the following is added to the Introduction:

“are commonly used to segment local behaviour along single trajectories (see Appendix 1 for comparisons).”

Also, we would like to mention that windowed analysis using mean squared displacements are conducted in Appendix 2—figure 1 (bottom row) showing the superior performance of our new method.

For further response on the biological relevance, see response to 3below.

3)It would be helpful if the authors could elaborate a bit further on how their fBm and Hurst exponents are useful to infer motion mechanism. The authors need to offer a bit more biological/cellular insight into their preceding findings on vesicular motion?

In order to elaborate further on biological insights and how fBm and Hurst exponents are useful to infer the motion mechanism, we introduce a new experimental dataset for SNX1 positive endosomes made by Anna Gavrilova (who is now also included as an author) and we add new biologically relevant analysis to the manuscript. To reflect this increased biological content, we have replaced the section ‘Trajectory analysis using DLFNN shows regime switching’ with ‘DLFNN analysis reveals differences in motile behavior of organelles in the endocytic pathway’. We also now include an introduction to the endocytic pathway (Introduction).

We tracked GFP-SNX1 tagged endosomes and we analyzed them in the same way as before to show a significant difference in the distribution of Hurst exponents (new data in Figure 4 and Figure 5). These results are intriguing, since both Rab5 and SNX1 are present on many of the same early endosomes, although in distinct domains (shown by confocal imaging in Figure 6—figure supplement 4). We now include example tracks and DLFNN analysis for a GFP-SNX1 endosome and a lysosome, for comparison (Figure 3—figure supplement 1 and Figure 3—figure supplement 2). We also include example videos for each organelle marker (Figure 6—figure supplement 1, Figure 6—figure supplement, Figure 6—figure supplement 3).

Importantly, we extend our analysis and biological relevance considerably by sub-dividing persistent segments of long runs into anterograde (outward, kinesin-driven) and retrograde transport (dynein-driven movement towards the cell center), as described in the Materials and methods section. These new results are presented in Figure 7 and Table 1. Altogether, this powerful combination of approaches reveals unexpected differences in behavior of the different endocytic compartments, even though they are using the same motor, dynein, for retrograde movement. Molecular mechanisms that might generate such distinct dynamics are discussed (subsection “DLFNN analysis reveals differences in motile behavior of organelles in the endo cytic pathway”). The results suggest that the manner in which these vesicles move is highly correlated with the different functions of vesicles on the endocytic pathway. This implies that directionality and the correlation between consecutive steps are important measurements in addition to the displacement, velocity and duration of movement.

Reviewer #1:The authors provide an algorithmic approach for evaluating cellular transport distinct from MSD based approaches. The useful feature is that presumably fewer points are needed for predictive value. This is important since photobleaching is a major limitation in particle tracking, the more information that can be extracted from limited data, the better.1.1) I feel this approach may be useful, but it is not clear how this compares to the current standard HMM analysis in a head to head matchup.

Please see response 2 to the essential revisions.

1.2) Also, it seems a bit cumbersome so anything they can do to make it user friendly would be appreciated.

We had previously made a GUI for the initial submission but was unable to make the GUI available on a public platform due to its size (∼1GB). However, after a long and tedious process of software engineering, we have been able to reduce the size of the GUI (∼100MB) and it is available online at https://zenodo.org/record/3613843#.XkPf2Wj7SUl. We have included this information in subsection “Software and code”:

“and a GUI is available on https://zenodo.org/record/3613843#.XkPf2Wj7SUl.”

Reviewer #2:The authors develop a useful Deep Learning Neural Network to classify heterogeneous single-particle motion from live-cell imaging data. They apply their model to infer fractional Brownian motion from both simulated and cellular data, and show that it out-performs competing approaches such as MSD-based averaging and exponent inference, accurately predicting Hurst exponents in as few as 7 steps. Their article is well written and presented.2.1) Training the neural networks on a fractional Brownian motion (fBm) model alone must effectively "bias" the inference procedure towards this model, acting analogous to a prior in Bayesian inference. Can the authors either explain or illustrate how training their model solely on the class of models they seek to infer, rather than a broader set of models, may bias their inference procedure? For example, could they perform a test of this potential bias directly by training the neural network by including other models that are not fBm, and show how this impacts their results, particularly in the case of cellular datasets?

Please see response 1 to the essential revisions.

2.2) In this vein, the authors cite HMM-based models that infer diffusion and directed motion from single-particle trajectories, but they do not compare their procedure with these methods. Some limited comparison would be helpful for the cellular datasets in particular in order to illustrate to the reader how/why application of the fBm to infer single-particle trajectory data from living systems is useful for inferring molecular/other mechanism of dynamical molecular motion in living systems.

Please see response 2 to the essential revisions

2.3) Related to this, it would also be helpful if the authors could elaborate a bit further on how their fBm and Hurst exponents are useful to infer motion mechanism, immediately before Conclusions, where the authors write:"This implies that the vesicles may have a biological mechanism to prioritise certain interactions within the complex cytoplasm, similar to how human dynamics are often heavy tailed and bursty Barabasi, (2005)."Given the rather large difference between human dynamics and molecular, organelle, or vesicular dynamics, could the authors offer a bit more biological/cellular insight into their preceding findings on vesicular motion?

Please see response 3 to the essential revisions where we further elaborate on the usefulness of fBm and Hurst exponents. The similarity between superdiffusive dynamics of vesicle transport and human dynamics is based on the origin of heavy tailed dynamics: most tasks which are performed (by humans or chemical reactions in biological/cellular context) being rapidly executed, whereas a few tasks (reactions which control directional switching of vesicles) experience very long waiting times due to complex signaling processes that lead to power-law distribution of uni-directional run lengths. A detailed elaboration of this idea is a work in progress.

2.4) For a wide application of this analysis tool in biology, can the authors provide a directly executable GUI software?

Please see response 1.2.

Reviewer #3:This paper describes an analysis tool for particle trajectory data. The authors used a deep learning feedforward neural network (DLFNN) to extract a stochastic Hurst exponent H(t) from a trajectory. They found that the neural network is a more sensitive method to characterize fractional Brownian motion (fBM) than previous statistical tools such as mean squared displacement (MSD), rescaled range, and sequential range methods. They applied this tool to analyze the trajectories of lysosome and endosomes in live cells. The topic is interesting, but the novelty and impact of the work are not very clear. Also, the software is based on Python, which may limit the application of the tool by a wide range of researchers in cell biology. A user-friendly, standalone software would be more helpful.3.1) As the authors mentioned in the Introduction, exponent estimation using neural networks has been already demonstrated. The authors claimed a novelty in that the local H(t) is used to segment single trajectories into persistent and anti-persistent sections. However, the manuscript is lacking comparison with the existing methods using hidden Markov models and rolling windowed analysis.

Please see response 2to Essential revisions and response 3.2.

3.2) The authors also claimed that "fBM with a stochastic Hurst exponent is a new intracellular transport model". However, this section is rather brief, and some notations are not clearly defined. I think stronger impact and novelty are required for publication in eLife.

We thank the reviewer 3 for this comment since it pushed us to develop further the mathematical aspects of the research. In fact, multifractional processes with random exponent as a theory already exists [Ayache and Taqqu, 2005]. However, there were no clear applications of this theory in experiments since there were no methods to measure the changing exponents for such a small number of points. To the authors’ knowledge, this paper is the first time that a changing Hurst exponent was detected, measured and applied in a way that was biologically meaningful. This became possible with the increasing capabilities of the neural network as an estimator.

Although estimating the anomalous exponent with a neural network was already demonstrated in [Bondarenko, Bugueva, and Dedok, 2016], our work presents a far more detailed evaluation of the neural network estimation for very short trajectories as well as demonstrating a biological application. To the authors’ knowledge, this is also the first time where the Hurst exponent estimation of the neural network has been characterized to such an extent, with comparisons to other statistical methods. Furthermore, we believe that this is also the first time that the Hurst exponent has been used to segment experimental trajectories, generating useful biological information. We have also shown that the Hurst exponent displays values of *H >* 0.5 when the windowed velocity of the endosome or lysosome is much greater than zero (displaying correlated directional motion) and *H <* 0.5 when the windowed velocity is close to zero, confirming that the Hurst exponent is a consistent measure for our experimental data.

In order to clarify the impact and the novelty of our work we change the last sentence of the Abstract to:

“By using this analysis, fractional Brownian motion with a stochastic Hurst exponent was used to interpret, for the first time, anomalous intracellular dynamics, revealing unexpected differences in behavior between closely related endocytic organelles.”

Also, we rewrite the subsection “‘fBm with a stochastic Hurst exponent is a new possible intracellular transport model’”:

“In our case, *H*(*t*) is itself a stochastic process and such a process has been considered theoretically [9]. This is the first application of such a theory to intracellular transport and opens a new avenue for characterizing vesicular movement. Furthermore, Figure 3 shows that the motion of a vesicle, *B_H_*(*t*), exhibits regime switching behaviour between persistent and anti-persistent states.”

Finally, we add the following to the end of the Conclusion:

“Finally, in addition to providing a new segmentation method of active and passive transport, this new technique distinguishes the difference in motility between lysosomes, Rab5 positive endosomes and SNX1 positive endosomes. The results suggest that the manner in which these vesicles move is highly linked with which endocytic pathway they are associated with especially when the motion is anti-persistent. This implies that directionality and the correlation between consecutive steps is important to measure in addition to the displacement, velocity and duration of movement. We hope that this type of analysis will allow discoveries in particle motility of a more refined nature and make applying anomalous transport theory more accessible to researchers in a wide variety of disciplines.”

3.3) For a wide application of this analysis tool in biology, can the authors provide a directly executable GUI software?

Please see response 1.2.

3.4) It is unclear whether users can apply the pre-trained model for a broad range of data with different time and spatial scales. Or do users have to train the neural network for their own data set?

Since the model requires differenced and normalized input values, we anticipate that it should be applicable to a wide range of data sets with no further training required. In order to clarify this, we add the following at the end of the subsection “DLFNN structure and training”:

“Since the model requires differenced and normalized input values, in theory, it should be applicable to a wide range of datasets. However, further testing must be done in order to confirm this.”

3.5) More detailed instruction for the training is required. In the subsection “DLFNN structure and training”, the last sentence needs more explanation.

We have clarified the training details in subsection “DLFNN structure and training”, where we specify the optimizer and the activation functions used for neural network training.

3.6) In Figure 1E, it is counterintuitive that the error (σ_H_) increases as the SNR increases. In subsection “The DLFNN is more accurate than established methods”, 'Gaussian noise with increasing signal-to-noise ratio" needs to be revised. Please use a different term for this parameter or revise the figure with a commonly used definition of SNR.

We thank the reviewer for pointing this out. Due to rearrangement of the Figure, 1E is now 1H. The x-axis label of Figure 1H has been revised to NoiseSignaland the text in subsection “The DLFNN is more accurate than established methods” has also been revised to ‘Figure 1H shows how the exponent estimation error increases when Gaussian noise with varying strength compared to the original signal is added to the fBm trajectories.’

3.7) It would be informative if the authors also show the Hurst exponent estimated from the TAMSD method in Figure 2.

We would like to draw the reviewer’s attention to Appendix 2—figure 1 where we have used a rolling window with TAMSD method to estimate the anomalous exponent.

3.8) Heavy-tailed distributions have been observed not only in human dynamics but also in many biological systems. Some of the relevant review and original articles are listed below.-Reynolds and Rhodes, (2009)-Ariel, et al., (2015)-Song et al., (2018)-Chen et al., (2015)

We thank reviewer 3 for pointing this out. We have added to subsection “fBm with a stochastic Hurst exponent is a new possible intracellular transport model”:

“ecological searching patterns [11], swarming bacteria [12] and…” and the Conclusion:

“…varying biological processes similar to ecological searching patterns [11], swarming bacteria [12] and…”